# Towards Maximizing the Representation Gap between In-Domain & Out-of-Distribution Examples

**Jay Nandy      Wynne Hsu      Mong Li Lee**
National University of Singapore
{jaynandy,whsu,leeml}@comp.nus.edu.sg

## Abstract

Among existing uncertainty estimation approaches, Dirichlet Prior Network (DPN) distinctly models different predictive uncertainty types. However, for in-domain examples with high data uncertainties among multiple classes, even a DPN model often produces indistinguishable representations from the out-of-distribution (OOD) examples, compromising their OOD detection performance. We address this shortcoming by proposing a novel loss function for DPN to maximize the *representation gap* between in-domain and OOD examples. Experimental results demonstrate that our proposed approach consistently improves OOD detection performance.

## 1   Introduction

Deep neural network (DNN) based models have achieved impeccable success to address various real-world tasks [1, 2, 3]. However, when these intelligent systems fail, they do not provide any explanation or warning. Predictive uncertainty estimation has emerged as an important research direction to inform users about possible wrong predictions and allow users to react in an informative manner, thus improving their reliability.

Predictive uncertainties of DNNs can come from three different sources: *model uncertainty, data uncertainty,* and *distributional uncertainty* [4, 5]. Model uncertainty or epistemic uncertainty captures the uncertainty in estimating the model parameters, conditioning on training data [4]. Data uncertainty (or aleatoric uncertainty) arises from the natural complexities of the underlying distribution, such as class overlap, label noise, homoscedastic and heteroscedastic noise, etc [4]. Distributional uncertainty or dataset shift arises due to the distributional mismatch between the training and test examples, that is, the test data is *out-of-distribution (OOD)* [6, 5].

It is useful to determine the sources of predictive uncertainties. In active learning, distributional uncertainty indicates that the classifier requires additional data for training. For real-world applications, where the cost of errors are high, such as in autonomous vehicles [7], medical diagnosis [8], financial, and legal fields [9], the source of uncertainty can allow manual intervention in an informed way.

Notable progress has been made for predictive uncertainty estimation. Bayesian neural network-based models conflate the distributional uncertainty through model uncertainty [10, 4, 11, 12, 13]. However, since the true posterior for their model parameters are intractable, their success depend on the nature of approximations. In contrast, non-Bayesian approaches can explicitly train the network in a multi-task fashion, incorporating both in-domain and OOD examples to produce sharp and uniform categorical predictions respectively [14, 15]. However, these approaches cannot robustly determine the source of predictive uncertainty [5]. In particular, the presence of high data uncertainty among multiple classes leads them to produce uniform categorical predictions for in-domain examples, often making them indistinguishable from the OOD examples.

Dirichlet Prior Network (DPN) separately models different uncertainty types by producing sharp uni-modal Dirichlet distributions for in-domain examples, and flat Dirichlet distributions for OOD

examples [5, 16]. It uses a loss function that explicitly incorporates Kullback-Leibler (KL)-divergence between the model output and a target Dirichlet with a pre-specified precision value. However, we show that for in-domain examples with high data uncertainties, their proposed loss function distributes the target precision values among the overlapping classes, leading to much flatter distributions. Hence, it often produces indistinguishable representations for such in-domain misclassified examples and OOD examples, compromising the OOD detection performance.

In this work, we propose an alternative approach for a DPN classifier that produces *sharp, multi-modal* Dirichlet distributions for OOD examples to maximize their *representation gap* from in-domain examples. We design a new loss function that separately models the mean and the precision of the output Dirichlet distributions by introducing a novel *explicit precision regularizer* along with the cross-entropy loss. Experimental results on several benchmark datasets demonstrate that our proposed approach achieves the best OOD detection performance.

## 2 Related Work

In the Bayesian neural network, the predictive uncertainty of a classification model is expressed in terms of data and model uncertainty [4]. Let $\mathcal{D}_{in} = \{\boldsymbol{x}_i, y_i\}_{i=1}^N \sim P_{in}(\boldsymbol{x}, y)$ where $\boldsymbol{x}$ and $y$ denotes the images and their corresponding class-labels, sampled from an underlying probability distribution $P_{in}(\boldsymbol{x}, y)$. Given an input $\boldsymbol{x}^*$, the data uncertainty, $p(\omega_c|\boldsymbol{x}^*, \boldsymbol{\theta})$ is the posterior distribution over class labels given the model parameters $\boldsymbol{\theta}$, while the model uncertainty, $p(\boldsymbol{\theta}|\mathcal{D}_{in})$ is the posterior distribution over parameters given the data, $\mathcal{D}_{in}$. Hence, the predictive uncertainty is given as:

$$p(\omega_c|\boldsymbol{x}^*, \mathcal{D}_{in}) = \int p(\omega_c|\boldsymbol{x}^*, \boldsymbol{\theta}) \ p(\boldsymbol{\theta}|\mathcal{D}_{in}) \ d\boldsymbol{\theta} \tag{1}$$

where $\omega_c$ is the representation for class $c$. We use the standard abbreviation for $p(y = \omega_c|\boldsymbol{x}^*, \mathcal{D}_{in})$ as $p(\omega_c|\boldsymbol{x}^*, \mathcal{D}_{in})$.

However, the true posterior of $p(\boldsymbol{\theta}|\mathcal{D}_{in})$ is intractable. Hence, we need approximation such as Monte-Carlo dropout (MCDP) [11], Langevin Dynamics [17], explicit ensembling [13]: $p(\omega_c|\boldsymbol{x}^*, \mathcal{D}_{in}) \approx \frac{1}{M} \sum_{m=1}^M p(\omega_c|\boldsymbol{x}^*, \boldsymbol{\theta}^{(m)})$. where, $\boldsymbol{\theta}^{(m)} \sim q(\boldsymbol{\theta})$ is sampled from an explicit or implicit variational approximation, $q(\boldsymbol{\theta})$ of the true posterior $p(\boldsymbol{\theta}|\mathcal{D}_{in})$. Each $p(\omega_c|\boldsymbol{x}^*, \boldsymbol{\theta}^{(m)})$ represents a categorical distribution, $\boldsymbol{\mu} = [\mu_1, \cdots, \mu_K] = [p(y = \omega_1), \cdots, p(y = \omega_K)]$ over class labels, given $\boldsymbol{x}^*$. Hence, the ensemble can be visualized as a collection of points on the probability simplex. For a confident prediction, it should be appeared sharply in one corner of the simplex. For an OOD example, it should be spread uniformly. We can determine the source of uncertainty in terms of the model uncertainty by measuring their spread. However, producing an ensemble distribution is computationally expensive. Further, it is difficult to control the desired behavior in practice [16]. Furthermore, for standard DNN models, with millions of parameters, it is even harder to find an appropriate prior distribution and the inference scheme to estimate the posterior distribution of the model.

Few recent works, such as Dirichlet prior network (DPN) [5, 16], evidential deep learning (EDL) [18] etc, attempt to emulate this behavior by placing a Dirichlet distribution as a prior, over the predictive categorical distribution. In particular, DPN framework [5, 16] significantly improves the OOD detection performance by explicitly incorporating OOD training examples, $\mathcal{D}_{out}$, as we elaborate in Section 3.

Non-Bayesian frameworks derive their measure of uncertainties using their predictive posteriors obtained from DNNs. Several works demonstrate that tweaking the input images using adversarial perturbations [19] can enhance the performance of a DNN for OOD detection [20, 21]. However, these approaches are sensitive to the tuning of parameters for each OOD distribution and difficult to apply for real-world applications. DeVries & Taylor (2018) [22] propose an auxiliary confidence estimation branch to derive OOD scores. Shalev et al. (2018) [23] use multiple semantic dense representations as the target label to train the OOD detection network. The works in [14, 15] also introduce multi-task loss, incorporating OOD data for training. Hein et al. (2019) [24] show that ReLU-networks lead to over-confident predictions even for samples that are far away from the in-domain distributions and propose methods to mitigate this problem [24, 25, 26]. While these models can identify the total predictive uncertainties, they cannot robustly determine whether the source of uncertainty is due to an in-domain data misclassification or an OOD example.

## 3 Dirichlet Prior Network

A DPN classification model directly parametrizes a Dirichlet distribution as a prior to the predictive categorical distribution over the probability simplex [5, 16]. It attempts to produce a sharp Dirichlet in a corner when it is confident in its predictions for in-domain examples (Fig 1a). For in-domain examples with high data uncertainty, it attempts to produce a sharp distribution in the middle of the simplex (Fig 1b). Note that, the probability densities for both Dirichlet distributions in Fig 1a and Fig 1b are concentrated in a single mode.

For OOD examples, an existing DPN produces a flat Dirichlet distribution to indicate high-order distributional uncertainty (see Fig 1c). However, we demonstrate that in the case of high data uncertainty among multiple classes, an existing DPN model also produces flatter Dirichlet distributions, leading to indistinguishable representations from OOD examples (see Section 4). Hence, we propose to produce sharp multi-modal Dirichlet distributions for OOD examples to increase their *"representation gap"* from in-domain examples, and improve the OOD detection performance (see Fig 1d). Note that, compared to Fig 1a or Fig 1b, the probability densities of both Dirichlet distributions in Fig 1c and Fig 1d are more scattered over the simplex. We can compute this *"diversity"* using measures, such as *"mutual information (MI)"*, to detect the OOD examples [5]. The predictive uncertainty of a DPN is given as:

Figure 1: Desired behavior of DPN classifiers to indicate different predictive uncertainty types.

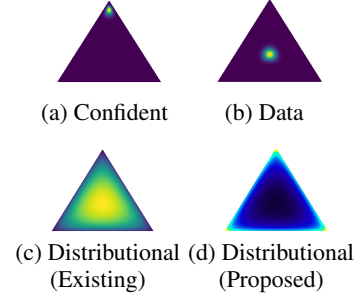

(a) Confident     (b) Data

(c) Distributional (d) Distributional
(Existing)      (Proposed)

$$p(\omega_c|\boldsymbol{x}^*, \mathcal{D}) = \int \int p(\omega_c|\boldsymbol{\mu})p(\boldsymbol{\mu}|\boldsymbol{x}^*, \boldsymbol{\theta})p(\boldsymbol{\theta}|\mathcal{D})d\boldsymbol{\mu}\, d\boldsymbol{\theta} \qquad (2)$$

where $\mathcal{D}$ denotes the training examples comprising both in-domain, $\mathcal{D}_{in}$, and OOD examples, $\mathcal{D}_{out}$.

Here, the data uncertainty, $p(\omega_c|\boldsymbol{\mu})$ is represented by point-estimate categorical distribution $\boldsymbol{\mu}$, while the distributional uncertainty is represented by using the distribution over the predictive categorical i.e. $p(\boldsymbol{\mu}|\boldsymbol{x}^*, \theta)$. A high model uncertainty, $p(\boldsymbol{\theta}|\mathcal{D})$ would induce a high variation in distributional uncertainty, leading to larger data uncertainty.

DPN is consistent with existing approaches where an additional term is incorporated for distributional uncertainty. Marginalization of $\boldsymbol{\mu}$ in Eqn. 2 produces Eqn. 1. Further, marginalizing $\boldsymbol{\theta}$ produces the expected estimation of data and distributional uncertainty given model uncertainty, i.e,

$$p(\omega_c|\boldsymbol{x}^*, \mathcal{D}) = \int p(\omega_c|\boldsymbol{\mu})\Big[\int p(\boldsymbol{\mu}|\boldsymbol{x}^*, \boldsymbol{\theta})p(\boldsymbol{\theta}|\mathcal{D})d\boldsymbol{\theta}\Big]d\boldsymbol{\mu} = \int p(\omega_c|\boldsymbol{\mu})p(\boldsymbol{\mu}|\boldsymbol{x}^*, \mathcal{D})d\boldsymbol{\mu} \qquad (3)$$

However, as in Eqn. 1, marginalization of $\boldsymbol{\theta}$ is not tractable. Hence, as before, we can introduce approximation techniques for $p(\boldsymbol{\theta}|\mathcal{D})$ to measure model uncertainty [11, 13]. However, the model uncertainty is *reducible* given large amount of training data [4]. Here, we only focus on the uncertainties introduced by the input examples and assume a dirac-delta approximation $\hat{\boldsymbol{\theta}}$ for $\boldsymbol{\theta}$ for the DPN models [5]: $p(\boldsymbol{\theta}|\mathcal{D}) = \delta(\boldsymbol{\theta} - \hat{\boldsymbol{\theta}}) \implies p(\boldsymbol{\mu}|\boldsymbol{x}^*, \mathcal{D}) \approx p(\boldsymbol{\mu}|\boldsymbol{x}^*, \hat{\boldsymbol{\theta}})$.

**Construction.** A Dirichlet distribution is parameterized using a vector, called the concentration parameters, $\boldsymbol{\alpha} = \{\alpha_1, \cdots, \alpha_K\}$, as: $Dir(\boldsymbol{\mu}|\boldsymbol{\alpha}) = \frac{\Gamma(\alpha_0)}{\prod_{c=1}^{K}\Gamma(\alpha_c)}\prod_{c=1}^{K}\mu_c^{\alpha_c-1}$, $\alpha_c > 0$, where, $\alpha_0 = \sum_{c=1}^{K}\alpha_c$) denotes its precision. A larger precision leads to a sharper Dirichlet distribution.

A DPN, $f_{\hat{\theta}}$ produces the concentration parameters, $\boldsymbol{\alpha} = \{\alpha_1, \cdots, \alpha_K\}$ corresponding to each class, i.e, $\boldsymbol{\alpha} = f_{\hat{\theta}}(\boldsymbol{x}^*)$. The posterior over class labels is given by the mean of the Dirichlet, i.e,

$$p(\omega_c|\boldsymbol{x}^*; \hat{\boldsymbol{\theta}}) = \int p(\omega_c|\boldsymbol{\mu})\ p(\boldsymbol{\mu}|\boldsymbol{x}^*; \hat{\boldsymbol{\theta}})\ d\boldsymbol{\mu} = \frac{\alpha_c}{\alpha_0}, \qquad \text{where,}\ \ p(\boldsymbol{\mu}|\boldsymbol{x}^*; \hat{\boldsymbol{\theta}}) = Dir(\boldsymbol{\mu}|\boldsymbol{\alpha}) \qquad (4)$$

A standard DNN with the softmax activation function can be represented as a DPN where the concentration parameters are $\alpha_c = e^{z_c(\boldsymbol{x}^*)}$; $z_c(\boldsymbol{x}^*)$ is the pre-softmax (logit) output corresponding to the class, $c$ for an input $\boldsymbol{x}^*$. The expected posterior probability of class label $\omega_c$ is given as:

$$p(\omega_c|\boldsymbol{x}^*; \hat{\boldsymbol{\theta}}) = \frac{\alpha_c}{\alpha_0} = \frac{e^{z_c(\boldsymbol{x}^*)}}{\sum_{c=1}^{K} e^{z_c(\boldsymbol{x}^*)}} \qquad (5)$$

However, the mean of the Dirichlet distribution is now *insensitive* to any arbitrary scaling of $\alpha_c$. Hence, while the standard *cross-entropy loss* efficiently models the mean of the Dirichlet distributions, it degrades the precision, $\alpha_0$.

Malinin & Gales (2018) [5] propose a forward KL (FKL) divergence loss that explicitly minimizes the KL divergence between the model and the given target Dirichlet distribution. Malinin & Gales (2019) [16] further propose a reverse KL (RKL) loss function that reverses the terms in the KL divergence to induce a uni-modal Dirichlet as the target distribution and improve their scalability for classification tasks with a larger number of classes. The RKL loss trains a DPN using both in-domain and OOD training examples in a multi-task fashion:

$$\mathcal{L}^{rkl}(\theta; \gamma, \boldsymbol{\beta}^y, \boldsymbol{\beta}^{out}) = \mathbb{E}_{P_{in}} \text{KL}[p(\boldsymbol{\mu}|\boldsymbol{x}, \boldsymbol{\theta})||Dir(\boldsymbol{\mu}|\boldsymbol{\beta}^y)] + \gamma \cdot \mathbb{E}_{P_{out}} \text{KL}[p(\boldsymbol{\mu}|\boldsymbol{x}, \boldsymbol{\theta})||Dir(\boldsymbol{\mu}|\boldsymbol{\beta}^{out})] \quad (6)$$

where $P_{in}$ and $P_{out}$ are the distribution for the in-domain and OOD training examples and $\boldsymbol{\beta}^y$ and $\boldsymbol{\beta}^{out}$ their hand-crafted target concentration parameters respectively.

## 4 Proposed Methodology

**Shortcomings of DPN using RKL loss.** We first demonstrate that the RKL loss function tends to produce flatter Dirichlet distributions for in-domain misclassified examples, compared to its confident predictions. We can decompose the reverse KL-divergence loss into two terms i.e *reverse cross entropy*, $\mathbb{E}_{P(\boldsymbol{\mu}|\boldsymbol{x}, \boldsymbol{\theta})}[-\ln Dir(\boldsymbol{\mu}|\overline{\boldsymbol{\beta}})]$ and *differential entropy*, $\mathcal{H}[p(\boldsymbol{\mu}|\boldsymbol{x}, \boldsymbol{\theta})]$, as shown in [16]:

$$\mathbb{E}_{\tilde{P}_T(\boldsymbol{x}, y)} \text{KL}\big[ p(\boldsymbol{\mu}|\boldsymbol{x}, \boldsymbol{\theta}) || Dir(\boldsymbol{\mu}|\boldsymbol{\beta}) \big] = \mathbb{E}_{\tilde{P}_T(\boldsymbol{x})}\Big[ \mathbb{E}_{P(\boldsymbol{\mu}|\boldsymbol{x}, \boldsymbol{\theta})}[-\ln Dir(\boldsymbol{\mu}|\overline{\boldsymbol{\beta}})] - \mathcal{H}\big[p(\boldsymbol{\mu}|\boldsymbol{x}, \boldsymbol{\theta})\big]\Big] \quad (7)$$

where $\boldsymbol{\beta} = \{\beta_1^{(c)}, \cdots, \beta_K^{(c)}\}$ represents their hand-crafted target concentration parameters, and $\overline{\boldsymbol{\beta}}$ represents the concentration parameter of the expected target Dirichlet with respect to the empirical training distribution $\tilde{P}_T$. In our analysis, we replace $\tilde{P}_T$ with the empirical distribution of in-domain training examples $\tilde{P}_{in}$ or OOD training examples $\tilde{P}_{out}$.

Differential entropy measures the sharpness of a continuous distribution. Minimizing $-\mathcal{H}[p(\boldsymbol{\mu}|\boldsymbol{x}, \boldsymbol{\theta})]$ always leads to produce a flatter distribution. Hence, we rely only on $\mathbb{E}_{P(\boldsymbol{\mu}|\boldsymbol{x}, \boldsymbol{\theta})}[-\ln Dir(\boldsymbol{\mu}|\overline{\boldsymbol{\beta}})]$ to produce sharper distributions. Malinin & Gales (2019) [16] choose the target concentration value for in-domain examples as: $(\beta + 1)$ for the correct class and 1 for the incorrect classes. Thus, we get:

$$\mathbb{E}_{\tilde{P}_T(\boldsymbol{x})}\Big[\mathbb{E}_{P(\boldsymbol{\mu}|\boldsymbol{x}, \boldsymbol{\theta})}[-\ln Dir(\boldsymbol{\mu}|\overline{\boldsymbol{\beta}})]\Big] = \mathbb{E}_{\tilde{P}_T(\boldsymbol{x})}\Big[ \sum_c \sum_k \tilde{p}(\omega_c|\boldsymbol{x})(\beta_k^{(c)} - 1)\big[\psi(\alpha_0) - \psi(\alpha_k)\big]\Big]$$
$$= \mathbb{E}_{\tilde{P}_T(\boldsymbol{x})}\Big[\beta\psi(\alpha_0) - \sum_c \beta\tilde{p}(\omega_c|\boldsymbol{x})\psi(\alpha_c)\Big] \quad (8)$$

where $\psi$ is the digamma function.

We can see in Eqn. 8, the reverse cross-entropy term maximizes $\psi(\alpha_c)$ for each class $c$ with the factor, $\beta\tilde{p}(\omega_c|\boldsymbol{x})$, and minimizes $\psi(\alpha_0)$ with the factor, $\beta$. For an in-domain example with confident prediction, it produces a sharp Dirichlet with a large concentration value for the correct class and very small concentration parameters $(<< 1)$ for the incorrect classes. However, for an input with high data uncertainty, $\beta$ is distributed among multiple classes according to $\tilde{p}(\omega_c|\boldsymbol{x})$. This leads to relatively smaller (but $\geq 1$) concentration parameters for all overlapping classes, producing a much flatter and diverse Dirichlet.

For example, let us consider two Dirichlet distributions, $Dir_1$ and $Dir_2$, with the same precision, $\alpha_0 = 102$, but different concentration parameters of $\{0.01, 0.01, 101.98\}$ (Fig. 2a) and $\{34, 34, 34\}$ (Fig. 2b). We measure the differential entropy (D.Ent) and mutual information (MI), which are maximized for flatter and diverse distributions respectively. We can see that $Dir_1$ produces much lower D.Ent and MI scores than $Dir_2$. It shows

Figure 2: Dirichlet distributions with the same precision but different concentration parameters.

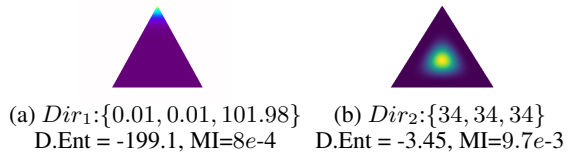

(a) $Dir_1$:$\{0.01, 0.01, 101.98\}$     (b) $Dir_2$:$\{34, 34, 34\}$
D.Ent = -199.1, MI=$8e$-4     D.Ent = -3.45, MI=$9.7e$-3

that $Dir_2$ is flatter and diverse than $Dir_1$, even with the same precision values. The differences in these scores become more significant in higher dimensions. As we consider the same example for a classification task with $K = 100$, D.Ent and MI would respectively produce $-9.9e3$ and $0.02$ for $Dir_1$ and $-370.5$ and $0.23$ for $Dir_2$. Further, in Section 4 (and in Table 9 (Appendix)), we show that DPN models also tend to produce lower precision values along with flatter and diverse Dirichlet distributions for misclassified examples.

This behavior is *not* desirable: since the RKL loss also trains the DPN to produce flatter and diverse Dirichlet distributions for OOD examples, it often leads to indistinguishable distributions for OOD examples in the boundary cases and in-domain misclassified examples. However, we can produce sharper Dirichlet distributions for OOD examples, as in Figure 1d, to maximize their representation gap from in-domain examples. For OOD training examples, we should choose identical values for target concentration parameters, $(\tau + 1)$ where $\tau > -1$, for all classes to produce uniform categorical posterior distributions. Using $(\tau + 1)$ for $\beta_k^{(c)}$ in Eqn. 8, we get the RKL loss for OOD examples as:

$$\mathbb{E}_{P_T(\boldsymbol{x})}\Big[\tau K \psi(\alpha_0) - \sum_c \tau \psi(\alpha_c) - \mathcal{H}\big[p(\boldsymbol{\mu}|\boldsymbol{x}, \boldsymbol{\theta})\big]\Big] \tag{9}$$

Malinin & Gales (2019) [16] choose $\tau$ to 0. This results the RKL loss to minimize $-\mathcal{H}[p(\boldsymbol{\mu}|\boldsymbol{x}, \boldsymbol{\theta})]$. Hence, the DPN produces flat Dirichlet distributions for OOD examples. In the following, we investigate the other choices of $\tau$.

Choosing $\tau > 0$ gives an objective function that minimizes the precision, $\alpha_0 (= \sum_{c=1}^{K} \alpha_c)$ of the output Dirichlet distribution while maximizing individual concentration parameters, $\alpha_c$ (Eq. 9). In contrast, choosing $\tau \in (-1, 0)$ maximizes $\alpha_0$ while minimizing $\alpha_c$'s. Hence, we can conclude that either choice of $\tau$ may lead to *uncontrolled values* for the concentration parameters for an OOD example. We also empirically verify this analysis in Appendix A.2.

**Explicit Precision Regularization.** We propose a new loss function for DPN classifiers that separately models the mean and precision of the output Dirichlet to achieve greater control over the desired behavior of a DPN classifier. We model the mean of the output Dirichlet using soft-max activation function and the cross-entropy loss, as in Eqn. 5, along with a novel *explicit precision regularizer*, that controls the precision.

Note that, we cannot choose the regularizer that directly maximizes the precision, $\alpha_0 = \sum_{c=1}^{K} \exp z_c(\boldsymbol{x})$ to control the output Dirichlet distributions. This is because the term, $\exp z_c(\boldsymbol{x})$ is unbounded. Hence, using the precision, $\alpha_0 = \sum_c \exp z_c(\boldsymbol{x})$ as the regularizer leads to large logit values for in-domain examples (see Figure 3(a)). However, it would make the cross-entropy loss term negligible, degrading the in-domain classification accuracy. Further, $\exp z_c(\boldsymbol{x})$ is not a symmetric function. Hence, it does not equally

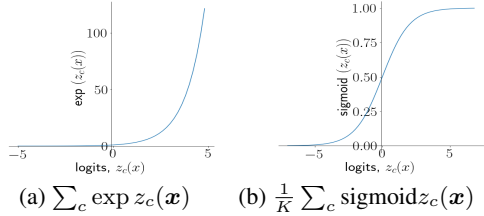

Figure 3: Growth of regularizers w.r.t logits.

(a) $\sum_c \exp z_c(\boldsymbol{x})$    (b) $\frac{1}{K} \sum_c \text{sigmoid} z_c(\boldsymbol{x})$

constrain the network to produce small fractional concentration parameters, i.e $\alpha_c = \exp z_c(\boldsymbol{x}) \rightarrow 0$, for OOD examples to produce the desirable multi-modal Dirichlet distributions (Figure 1d). Moreover, in practice, the choice of $\sum_c \exp z_c(\boldsymbol{x})$ leads the training loss to NaN.

In contrast, by limiting logits, $z_c$ to values that are, for example, approximately 5 for in-domain examples, and $-5$ for OOD examples, we would have the desirable sharp uni-modal and multi-modal Dirichlet distributions respectively, maximizing their representation gaps (see Figure 1). Beyond these values, the cross-entropy loss should become the dominant term in the loss function to improved the in-domain classification accuracy.

Hence, we propose a logistic-sigmoid approximation to *individually control* the concentration parameters using $\frac{1}{K} \sum_{c=1}^{K} \text{sigmoid}(z_c(\boldsymbol{x}))$ as the regularizer to control the spread of the output Dirichlet distributions. This regularizer is applied alongside the cross-entropy loss on the soft-max outputs. The use of logistic-sigmoid function satisfies this condition by providing an implicit upper and lower bounds on the individual concentration parameters for both in-domain and OOD examples (see Figure 3(b)). For interval $(-\infty, \epsilon)$, when $\epsilon$ is close to 0, the approximation error is low. While for the interval $[\epsilon, \infty)$, it offers the desired behavior of monotonically increasing function, similar to the exponential function, however within a finite boundary.

The proposed loss function for the in-domain training examples is given as:

$$\mathcal{L}_{in}(\boldsymbol{\theta}, \lambda_{in}) := \mathbb{E}_{P_{in}(\boldsymbol{x}, y)}\Big[-\log p(y|\boldsymbol{x}, \boldsymbol{\theta}) - \frac{\lambda_{in}}{K} \sum_{c=1}^{K} \text{sigmoid}(z_c(\boldsymbol{x}))\Big] \tag{10}$$

Similarly, for OOD training examples, we can control precision values using the loss function as:

$$\mathcal{L}_{out}(\boldsymbol{\theta}, \lambda_{out}) := \mathbb{E}_{P_{out}(\boldsymbol{x}, y)}\Big[\mathcal{H}_{ce}(\mathcal{U}; p(y|\boldsymbol{x}, \boldsymbol{\theta})) - \frac{\lambda_{out}}{K} \sum_{c=1}^{K} \text{sigmoid}(z_c(\boldsymbol{x}))\Big] \tag{11}$$

where $\mathcal{H}_{ce}$ is the cross-entropy function. $\mathcal{U}$ is the uniform distribution over the class labels. $\lambda_{in}$ and $\lambda_{out}$ are user-defined hyper-parameters for the regularization terms to control the precision of the output distributions. We train the DPN in a multi-task fashion using the overall loss function as:

$$\min_{\boldsymbol{\theta}} \mathcal{L}(\boldsymbol{\theta}; \gamma, \lambda_{in}, \lambda_{out}) = \mathcal{L}_{in}(\boldsymbol{\theta}, \lambda_{in}) + \gamma \mathcal{L}_{out}(\boldsymbol{\theta}, \lambda_{out}) \tag{12}$$

where $\gamma > 0$ balances between the loss values for in-domain examples and OOD examples. We now analyze the proposed regularizer by taking expectation with respect to the empirical distribution, $\tilde{P}_T$:

$$\mathbb{E}_{\tilde{P}_T(\boldsymbol{x},y)}\Big[ -\frac{\lambda_T}{K}\sum_{c=1}^{K}\text{sigmoid}(z_c(\boldsymbol{x}))\Big] = \mathbb{E}_{P_T(\boldsymbol{x})}\Big[ -\frac{\lambda_T}{K}\sum_{k=1}^{K}p(y=\omega_k|\boldsymbol{x})\Big[\sum_{c=1}^{K}\text{sigmoid}(z_c(\boldsymbol{x}))\Big]\Big]$$

$$= \mathbb{E}_{P_T(\boldsymbol{x})}\Big[ -\frac{\lambda_T}{K}\sum_{c=1}^{K}\text{sigmoid}(z_c(\boldsymbol{x}))\Big] \tag{13}$$

where $\lambda_T$ and $\tilde{P}_T$ should be replaced to $\lambda_{in}$ and $\tilde{P}_{in}$ respectively for in-domain examples, and $\lambda_{out}$ and $\tilde{P}_{out}$ respectively for OOD examples.

Hence, by choosing $\lambda_{in} > 0$ for in-domain examples, our regularizer imposes the network to maximize $\text{sigmoid}(z_c(\boldsymbol{x}))$ irrespective of the class-labels. However, for confidently predicted examples, the cross-entropy loss ensures to maximize the logit value for the correct class. In contrast, in the presence of high data uncertainty, the cross-entropy loss produces a multi-modal categorical distribution over the overlapping classes. Hence, as before, it leads to producing a flatter distribution for misclassified examples. Now, by choosing $\lambda_{in} > \lambda_{out} > 0$, we also enforce the network to produce a flatter distribution with $\alpha_c = \exp z_c(\boldsymbol{x}^*) \geq 1$ for an OOD example $\boldsymbol{x}^*$. Hence, the DPN will produce indistinguishable representations for an in-domain example with high data uncertainty as an OOD example, similar to the RKL loss (Eq. 7).

However, now we can address this problem by choosing $\lambda_{out} < 0$. It enforces the DPN to produce negative values for $z_c(\boldsymbol{x}^*)$ and thus *fractional* values for $\alpha_c$'s for OOD examples. This leads the probability densities to be moved across the edges of the simplex to produce extremely sharp multi-modal distributions, solving the original problem described in the beginning.

Figure 4: Dirichlet distributions with different precision.

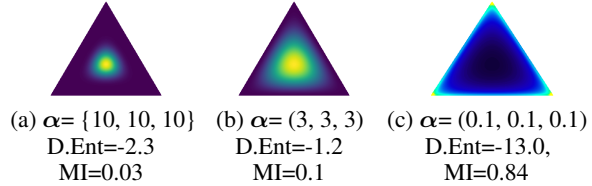

(a) $\boldsymbol{\alpha}=\{10, 10, 10\}$  (b) $\boldsymbol{\alpha}=(3, 3, 3)$  (c) $\boldsymbol{\alpha}=(0.1, 0.1, 0.1)$
D.Ent=-2.3    D.Ent=-1.2    D.Ent=-13.0,
MI=0.03       MI=0.1        MI=0.84

For example, let a DPN with $\lambda_{in}, \lambda_{out} > 0$ represents a misclassified example (Fig. 4a) and an OOD example (Fig. 4b). We can see that their representations are very similar, even if their concentration parameters are different. In contrast, our DPN with $\lambda_{in} > 0, \lambda_{out} < 0$ leads to a sharp, multi-modal Dirichlet as in Fig. 4(c) for an OOD, maximizing the *representation gap* from Fig. 4(a). We can confirm this by observing their *mutual information (MI)* scores. However, the choice of $\lambda_{in} = 0$ and $\lambda_{out} < 0$ does not enforce these properties (see ablation study in Appendix A.1).

The overall loss function in Eqn. 12 requires training samples from both in-domain distribution and OOD. Here, we select a different real-world dataset as our OOD training examples. It is more feasible in practice and performs better than the artificially generated OOD examples [15, 14].

## 5  Experimental Study

We conduct two sets of experiments: First, we experiment on a synthetic dataset. Next, we present a comparative study on a range of image classification tasks. [1] [2]

### 5.1  Synthetic Dataset

We construct a simple synthetic dataset with three overlapping classes to study the characteristics of different DPN models. We sample the in-domain training instances from three different overlapping

Figure 5: Uncertainty measures of different data-points for DPN$^+$ (top row) and DPN$^-$ (bottom row).

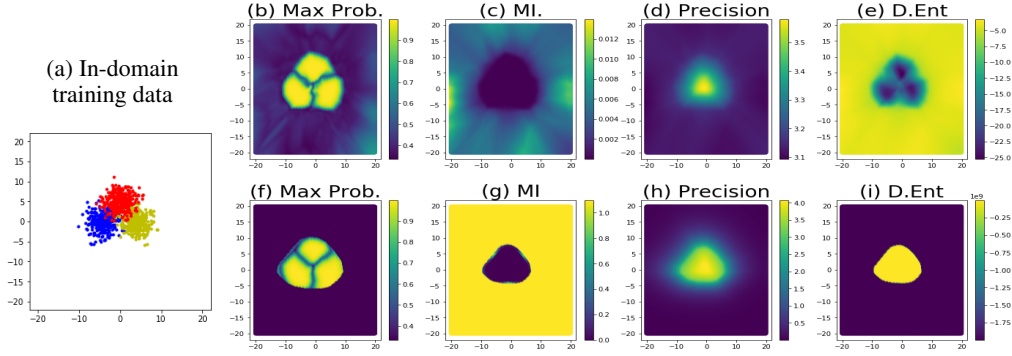

isotropic Gaussian distributions, to obtain these overlapping classes, as shown in Figure 5a. We demonstrate the results of our DPN models with $\lambda_{in} > 0$ and both positive and negative values for $\lambda_{out}$, are denoted as DPN$^+$ and DPN$^-$ respectively. See Appendix B.1 for additional details on experimental setup, hyper-parameters and the results using RKL loss function [16].

A *total predictive uncertainty* measure is derived from the expected predictive categorical distribution, $p(\omega_c|\boldsymbol{x}^*, D)$ i.e by marginalizing $\boldsymbol{\mu}$ and $\boldsymbol{\theta}$ in Eq. 2. Maximum probability in the expected predictive categorical distribution is a popular total uncertainty measure: $max\mathcal{P} = \max_c p(\omega_c|\boldsymbol{x}^*, D)$.

Fig 5b and Fig 5f show the Max.P uncertainty measures for different data points for DPN$^+$ and DPN$^-$ respectively. We can see that DPN$^-$ appropriately interpolates the concentration parameters to maximize the margin on the boundary of the in-domain and OOD regions, leading to improved OOD detection performance even using total uncertainty measures. However, since the predicted distributions are obtained by marginalizing $\boldsymbol{\mu}$ (Eqn. 2), the total uncertainty measures fail to robustly distinguish the OOD examples from the misclassified examples. As we can see in both Fig 5b and Fig 5f that the Max.P produces lower scores in the class overlapping regions. Since the non-Bayesian models only relies on the total uncertainty measures, they are unable to reliably distinguish between data and distributional uncertainty [5].

A DPN can address this limitation by computing the *mutual information (MI)* between $y$ and $\boldsymbol{\mu}$ i.e $\mathcal{I}[y, \boldsymbol{\mu}|\boldsymbol{x}^*, \hat{\boldsymbol{\theta}}]$. We can also measure the *expected pairwise KL divergence (EPKL)* between the pairs of independent "categorical distribution" samples from the Dirichlet [27]. For a Dirichlet distribution, EPKL is simplified to $\frac{K-1}{\alpha_0}$, where $\alpha_0$ is the precision [27]. Since a DPN also produces smaller precision values for OOD examples, we can directly view the *precision* as a distributional uncertainty measure. Note that, both EPKL and *precision (or inverse-EPKL)* leads to the same OOD detection performance as they produce the same relative uncertainty scores (in reverse order) for a given set of test examples. In Fig 5c and Fig 5d, we observe that our DPN$^+$ model successfully distinguishes the OOD examples using the mutual information and the precision measures respectively. However, our DPN$^-$ model clearly demonstrates its superiority by producing sharper and significant differences of uncertainty scores for OOD examples, compared to the in-domain examples (Fig 5g and Fig 5h).

*Differential entropy (D.Ent)*, $\mathcal{H}[p(\boldsymbol{\mu}|\boldsymbol{x}^*, \hat{\boldsymbol{\theta}})]$, that maximizes for sharp Dirichlet distributions, is also used as a distributional uncertainty measure [5, 27]. However, unlike other DPN models, our DPN$^-$ behaves differently to produce sharp multi-modal Dirichlet distributions for OOD examples. Hence, D.Ent also behaves in an *inverted manner*, compared to the other DPN models. As we can see in Fig 5e, DPN$^+$ produces higher D.Ent values for OOD examples. In contrast, DPN$^-$ produces *large negative* D.Ent scores for OOD examples, indicating that it often produces even sharper Dirichlet for OOD examples, than the confidently predicted examples (Fig 5i).

## 5.2   Benchmark Image Classification Datasets

Next, we carry out experiments on CIFAR-10 and CIFAR-100 [28] and TinyImageNet [29]. We train the C10 classifiers by using CIFAR-10 training images as in-domain data and CIFAR-100 training images as OOD data. C100 classifiers are trained by using CIFAR-100 training images as in-domain and CIFAR-10 training images as OOD. For the TIM classifier, we use the TinyImageNet images as in-domain training data and ImageNet-25K images as OOD training data. ImageNet-25K is obtained

by randomly selecting $25,000$ images from the ImageNet dataset [30]. We use the VGG-16 network for these tasks [1]. Here, we study the performance our DPN models with $\lambda_{in} > 0$ and both $\lambda_{out} > 0$ and $\lambda_{out} < 0$, are denoted as $DPN^+$ and $DPN^-$ respectively. See Appendix A for additional ablation studies. We compare our models with the standard DNN [31], Bayesian MCDP [11], Deep Ensemble (DE) [13], non-Bayesian OE [15] and the existing $DPN_{rev}$ model [16]. For Bayesian MCDP and DE, we can compute the mutual information (MI). However, we cannot compute the precision or D.Ent for them. For the non-Bayesian models, MI, precision, and D.Ent are not defined. See Appendix B.2 for our experimental details along with additional discussions.

We present the performances of our models for *OOD detection* and *misclassification detection* tasks. Note that the in-domain and OOD test examples are kept separately from the training examples, as in a real-world scenario (see Table 7(Appendix)). For OOD detection, we choose the OOD examples as the 'positive' class and in-domain examples as the 'negative' class. For misclassification detection, we consider the misclassified examples as the 'positive' class and correctly classified examples as the 'negative' class. Here, we use *area under the receiver operating characteristic (AUROC)* metric [31]. We present the results using Max.P, MI, $\alpha_0$ (or inverse-EPKL), and D.Ent. We report the (mean $\pm$ standard deviation) of three different models. We provide additional results including classification accuracy, and performance on a wide range of OOD datasets along with area under the precision-recall curve (AUPR) metric and entropy measure in Appendix D (Table 9-13).

Table 1: AUROC scores for OOD detection (mean $\pm$ standard deviation of 3 runs). Refer to Table 11-13 (Appendix) for AUPR scores and results on additional OOD test sets.

| | OOD | Tiny [29] | | | | STL-10 [32] | | | | LSUN [33] | | | |
|---|---|---|---|---|---|---|---|---|---|---|---|---|---|
| | | Max.P | MI | $\alpha_0$ | D.Ent | Max.P | MI | $\alpha_0$ | D.Ent | Max.P | MI | $\alpha_0$ | D.Ent |
| C10 | Baseline | 88.9±0.0 | - | - | - | 75.9±0.0 | - | - | - | 90.3±0.0 | - | - | - |
| | MCDP | 88.7±0.1 | 88.1±0.1 | - | - | 76.2±0.0 | 76.0±0.0 | - | - | 90.6±0.0 | 90.2±0.0 | - | - |
| | DE | 88.9±NA | 87.8±NA | - | - | 76.0±NA | 75.6±NA | - | - | 90.3±NA | 89.7±NA | - | - |
| | OE | 98.2±0.1 | - | - | - | 81.4±1.2 | - | - | - | 98.4±0.3 | - | - | - |
| | DPN$_{rev}$ | 97.5±0.5 | 97.8±0.4 | 97.8±0.4 | 97.7±0.4 | 81.6±1.7 | 82.2±1.7 | 82.2±1.6 | 81.9±1.7 | 98.5±0.4 | 98.7±0.3 | 98.7±0.3 | 98.7±0.3 |
| | DPN$^+$ | 98.0±0.2 | 98.0±0.2 | 98.0±0.2 | 98.0±0.2 | 81.6±1.4 | 81.8±1.2 | 81.8±1.2 | 81.8±1.2 | 98.2±0.3 | 98.3±0.4 | 98.3±0.4 | 98.3±0.4 |
| | DPN$^-$ | **99.0**±0.1 | **99.0**±0.1 | 97.7±0.1 | 6.0±0.3 | 84.7±0.4 | **85.3**±0.5 | 84.9±0.5 | 34.6±0.4 | 99.2±0.1 | **99.3**±0.0 | 98.1±0.1 | 5.0±0.2 |

| | OOD | Tiny [29] | | | | STL-10 [32] | | | | LSUN [33] | | | |
|---|---|---|---|---|---|---|---|---|---|---|---|---|---|
| | | Max.P | MI | $\alpha_0$ | D.Ent | Max.P | MI | $\alpha_0$ | D.Ent | Max.P | MI | $\alpha_0$ | D.Ent |
| C100 | Baseline | 68.8±0.2 | - | - | - | 69.6±0.0 | - | - | - | 72.5±0.0 | - | - | - |
| | MCDP | 69.7±0.3 | 70.6±0.3 | - | - | 70.7±0.1 | 71.6±0.2 | - | - | 74.5±0.1 | 75.9±0.2 | - | - |
| | DE | 68.9±NA | 69.6±NA | - | - | 69.6±NA | 70.2±NA | - | - | 72.6±NA | 73.4±NA | - | - |
| | OE | 89.5±1.0 | - | - | - | 91.2±0.7 | - | - | - | 92.2±0.9 | - | - | - |
| | DPN$_{rev}$ | 81.2±0.2 | 83.8±0.1 | 83.8±0.1 | 83.5±0.1 | 87.2±0.1 | 89.3±0.1 | 89.3±0.1 | 89.0±0.1 | 86.7±0.0 | 89.3±0.1 | 89.3±0.1 | 88.9±0.1 |
| | DPN$^+$ | 85.9±0.3 | 92.2±0.1 | 92.2±0.1 | 92.3±0.1 | 89.1±0.2 | 95.0±0.0 | 95.0±0.0 | 94.8±0.0 | 90.3±0.3 | 95.0±0.1 | 95.0±0.1 | 95.0±0.1 |
| | DPN$^-$ | 89.2±0.1 | **94.5**±0.1 | **94.5**±0.1 | 38.1±0.5 | 92.8±0.1 | **96.8**±0.1 | **96.8**±0.1 | 25.4±0.4 | 92.8±0.1 | **96.5**±0.1 | **96.5**±0.1 | 31.5±0.4 |

| | OOD | CIFAR-10 [28] | | | | CIFAR-100 [28] | | | | Textures [34] | | | |
|---|---|---|---|---|---|---|---|---|---|---|---|---|---|
| | | Max.P | MI | $\alpha_0$ | D.Ent | Max.P | MI | $\alpha_0$ | D.Ent | Max.P | MI | $\alpha_0$ | D.Ent |
| TIM | Baseline | 76.9±0.2 | - | - | - | 73.6±0.2 | - | - | - | 70.9±0.2 | - | - | - |
| | MCDP | 77.4±0.1 | 77.5±0.2 | - | - | 74.0±0.2 | 73.6±0.2 | - | - | 70.3±0.2 | 63.6±0.2 | - | - |
| | DE | 76.9±NA | 77.7±NA | - | - | 73.7±NA | 75.3±NA | - | - | 71.1±NA | 76.2±NA | - | - |
| | OE | 91.3±0.4 | - | - | - | 89.5±0.5 | - | - | - | 95.8±0.3 | - | - | - |
| | DPN$_{rev}$ | 85.4±0.7 | 82.8±1.4 | 81.9±1.6 | 85.6±0.9 | 84.2±0.8 | 82.5±1.4 | 81.7±1.6 | 85.0±0.9 | 90.9±0.3 | 91.2±0.6 | 90.6±0.6 | 92.6±0.3 |
| | DPN$^+$ | 99.2±0.0 | 99.7±0.0 | 99.7±0.0 | 99.6±0.0 | 98.8±0.0 | 99.5±0.0 | 99.5±0.0 | 99.4±0.0 | 96.5±0.1 | 98.4±0.0 | 98.4±0.0 | 98.2±0.0 |
| | DPN$^-$ | 99.7±0.0 | **99.9**±0.0 | **99.9**±0.0 | 3.5±0.1 | 98.7±0.1 | **99.6**±0.0 | **99.6**±0.0 | 7.5±0.2 | 95.8±0.1 | **98.7**±0.1 | **98.7**±0.1 | 19.3±0.4 |

Tables 1 shows the performance of C10, C100 and TIM classifiers for *OOD detection* task. We observe that our $DPN^-$ models consistently outperform the other models using mutual information (MI) measure. Our $DPN^-$ models produce sharp multi-modal Dirichlet distributions for OOD examples, leading to higher MI scores compared to the in-domain examples. In contrast, $DPN^-$ models produce sharp Dirichlet distributions for both in-domain confident predictions and OOD examples. Hence, we cannot use D.Ent to distinguish them. However, in Table 3, we show that we can combine D.Ent with a total uncertainty measure to distinguish the in-domain and OOD examples.

Table 2: AUROC scores for misclassified image detection. Refer to Table 9 (Appendix) for additional results by using AUPR scores along with in-domain classification accuracy.

| | C10 | | | | C100 | | | | TIM | | | |
|---|---|---|---|---|---|---|---|---|---|---|---|---|
| | Max.P | MI | $\alpha_0$ | D.Ent | Max.P | MI | $\alpha_0$ | D.Ent | Max.P | MI | $\alpha_0$ | D.Ent |
| Baseline | 93.3±0.1 | - | - | - | 86.8±0.1 | - | - | - | 86.7±0.0 | - | - | - |
| MCDP | **93.6**±0.2 | 93.2±0.1 | - | - | **87.2**±0.0 | 83.3±0.3 | - | - | 86.6±0.1 | 83.3±0.3 | - | - |
| DE | 93.5±NA | 92.7±NA | - | - | 87.0±NA | 83.4±NA | - | - | **86.8**±NA | 83.3±NA | - | - |
| OE | 92.0±0.0 | - | - | - | 86.9±0.0 | - | - | - | 85.9±0.2 | - | - | - |
| DPN$_{rev}$ | 89.6±0.1 | 88.7±0.2 | 88.7±0.2 | 89.0±0.2 | 79.3±0.1 | 73.5±0.1 | 73.1±0.1 | 75.7±0.1 | 81.9±0.3 | 72.2±0.7 | 70.2±0.9 | 78.3±0.3 |
| DPN$^+$ | 92.2±0.3 | 90.3±0.1 | 90.3±0.1 | 90.5±0.2 | 86.5±0.1 | 81.2±0.0 | 81.3±0.0 | 81.9±0.1 | 85.7±0.2 | 78.3±0.4 | 78.7±0.5 | 79.7±0.2 |
| DPN$^-$ | 92.6±0.1 | 89.9±0.0 | 89.9±0.0 | 66.2±0.7 | 86.4±0.1 | 82.3±0.0 | 82.3±0.0 | 81.7±0.1 | 85.4±0.1 | 79.1±0.5 | 79.4±0.4 | 79.9±0.2 |

Table 2 presents the results for in-domain *misclassification detection*. We can see that our proposed $DPN^-$ models achieve comparable performance as the existing methods. It is interesting to note that, all the DPN models produce comparable AUROC scores using the distributional uncertainty measures as the total uncertainty measure. This supports our assertion that *in the presence of data uncertainty, DPN models tend to produce flatter and diverse Dirichlet distributions with smaller precisions for misclassified examples, compared to the confident predictions.* Hence, we should aim to produce sharp multi-modal Dirichlet distributions for OOD examples to keep them distinguishable from the in-domain examples. In Appendix B.3, we also demonstrate that our $DPN^-$ models improve calibration performance for classification [35, 15].

AUROC (and also AUPR) scores, in Table 1, only provide a relative measure of separation, while providing no information about how different these uncertainty values are for in-domain and OOD examples [36]. An OOD detection model should aim to maximize the margin of uncertainty values produced for the OOD examples from the in-domain examples to separate them efficiently. We can measure the "gap" of uncertainty values for in-domain and OOD examples by measuring the divergence of their distributions for uncertainty values produced by the models.

$DPN^-$ models produce sharp Dirichlet distributions for both OOD examples and in-domain confident predictions, while flat Dirichlet distributions for misclassified examples. Hence, we can combine D.Ent along with a total uncertainty measure, Max.P, to robustly distinguish them. Note that D.Ent ranges from $(-\infty, \infty)$ and often produce large negative values. Hence, we use $\log(-$D.Ent$)$. We separately consider Max.P and $\log(-$D.Ent$)$ scores for the in-domain correctly classified, misclassified, and OOD examples. We fit three different bivariate Gaussian distributions on these values. We then compute the KL-divergence of the distribu-

Figure 6: Illustrating the distribution of uncertainty values for $DPN^-$ and other DPN models. We normalize the scores for better visualization.

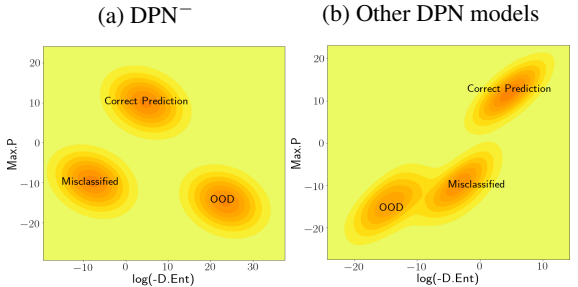

tions of uncertainty values for in-domain confidently predicted and misclassified examples to the OOD examples to measure their separability. Figure 6 illustrates the desired behavior of our $DPN^-$, compared to the other DPN models.

Table 3: KL-divergence scores from the distribution of uncertainty values of missclassified and correctly predicted examples to the OOD examples. See Table 10 (Appendix) for additional results.

|  | C10 | | | | C100 | | | | TIM | | | |
|---|---|---|---|---|---|---|---|---|---|---|---|---|
| OOD | Tiny [29] | | LSUN [33] | | Tiny [29] | | LSUN [33] | | CIFAR-10 [28] | | CIFAR-100 [28] | |
|  | Miss | Correct | Miss | Correct | Miss | Correct | Miss | Correct | Miss | Correct | Miss | Correct |
| $DPN_{rev}$ | $1.4_{\pm0.2}$ | $12.6_{\pm0.8}$ | $2.1_{\pm0.3}$ | $13.9_{\pm1.0}$ | $1.8_{\pm0.0}$ | $6.6_{\pm0.1}$ | $2.6_{\pm0.0}$ | $8.6_{\pm0.1}$ | $9.0_{\pm0.6}$ | $9.2_{\pm1.0}$ | $2.3_{\pm0.4}$ | $3.2_{\pm0.5}$ |
| $DPN^+$ | $1.5_{\pm0.2}$ | $12.1_{\pm1.9}$ | $2.0_{\pm0.2}$ | $12.7_{\pm2.0}$ | $1.8_{\pm0.0}$ | $7.2_{\pm0.2}$ | $2.6_{\pm0.0}$ | $9.1_{\pm0.2}$ | $21.1_{\pm3.4}$ | $27.1_{\pm3.9}$ | $17.7_{\pm3.0}$ | $23.5_{\pm3.5}$ |
| $DPN^-$ | $\mathbf{2.1_{\pm0.1}}$ | $\mathbf{20.7_{\pm1.2}}$ | $\mathbf{2.4_{\pm0.1}}$ | $\mathbf{22.5_{\pm1.3}}$ | $\mathbf{44.5_{\pm2.8}}$ | $\mathbf{244.2_{\pm25.2}}$ | $\mathbf{49.9_{\pm3.5}}$ | $\mathbf{272.0_{\pm29.3}}$ | $\mathbf{729.8_{\pm12.4}}$ | $\mathbf{1360.4_{\pm94.5}}$ | $\mathbf{664.9_{\pm11.6}}$ | $\mathbf{1241.4_{\pm79.8}}$ |

In Table 3, the significantly higher KL-divergence for our $DPN^-$ models indicate that by combining Max.P with D.Ent measure, we can easily distinguish the OOD examples from the in-domain examples. Further, as we consider classification tasks with a larger number of classes, our $DPN^-$ produces more number of fractional concentration parameters for each class for the OOD examples. It further increases the $\log(-$D.Ent$)$ values, leading to maximizing the "gaps" between OOD examples from both in-domain confident predictions as well as misclassified examples.

## 6   Conclusion

The existing formulation for DPN models often lead to indistinguishable representations between in-domain examples with high data uncertainty among multiple classes and OOD examples. In this work, we have proposed a novel loss function for DPN models that maximizes the representation gap between in-domain and OOD examples. Experiments on benchmark datasets demonstrate that our proposed approach effectively distinguishes the distributional uncertainty from other uncertainty types and outperforms the existing OOD detection models.

# 7 Broader Impact

Despite the impeccable success of deep neural network (DNN)-based models in various real-world applications, they often produce incorrect predictions without proving any warning for the users. It raises the question of how much can we trust these models and whether it is safe to use them for sensitive real-world applications such as medical diagnosis, self-driving cars or to make financial decisions.

In this paper, we aim to robustly identify the source of uncertainty in the prediction of a DNN based model for classification tasks. Identifying the source of uncertainty in the prediction would allow manual intervention in an informed way and make an AI system more reliable for real-world applications. In particular, we address a shortcoming of the existing techniques and propose a novel solution to improve the detection of anomalous out-of-distribution examples for a classification model.

## Acknowledgement

This research is supported by the National Research Foundation, Singapore under its AI Singapore Programme (AISG Award No: AISG-GC-2019-001). Any opinions, findings and conclusions or recommendations expressed in this material are those of the authors and do not reflect the views of National Research Foundation, Singapore

## Footnotes

[1]Please refer to Appendix for additional results and ablation studies.

[2]Code Link: https://github.com/jayjaynandy/maximize-representation-gap

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
