[Supplementary Material]

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

 | $88.9_{\pm0.0}$ | - | - | - | $75.9_{\pm0.0}$ | - | - | - | $90.3_{\pm0.0}$ | - | - | - |
| MCDP | $88.7_{\pm0.1}$ | $88.1_{\pm0.1}$ | - | - | $76.2_{\pm0.0}$ | $76.0_{\pm0.0}$ | - | - | $90.6_{\pm0.0}$ | $90.2_{\pm0.0}$ | - | - |
| DE | $88.9_{\pm NA}$ | $87.8_{\pm NA}$ | - | - | $76.0_{\pm NA}$ | $75.6_{\pm NA}$ | - | - | $90.3_{\pm NA}$ | $89.7_{\pm NA}$ | - | - |
| OE | $98.2_{\pm0.1}$ | - | - | - | $81.4_{\pm1.2}$ | - | - | - | $98.4_{\pm0.3}$ | - | - | - |
| $DPN_{rev}$ | $97.5_{\pm0.5}$ | $97.8_{\pm0.4}$ | $97.8_{\pm0.4}$ | $97.7_{\pm0.4}$ | $81.6_{\pm1.7}$ | $82.2_{\pm1.7}$ | $82.2_{\pm1.6}$ | $81.9_{\pm1.7}$ | $98.5_{\pm0.4}$ | $98.7_{\pm0.3}$ | $98.7_{\pm0.3}$ | $98.7_{\pm0.3}$ |
| $DPN^+$ | $98.0_{\pm0.2}$ | $98.0_{\pm0.2}$ | $98.0_{\pm0.2}$ | $98.0_{\pm0.2}$ | $81.6_{\pm1.4}$ | $81.8_{\pm1.2}$ | $81.8_{\pm1.2}$ | $81.8_{\pm1.2}$ | $98.2_{\pm0.3}$ | $98.3_{\pm0.4}$ | $98.3_{\pm0.4}$ | $98.3_{\pm0.4}$ |
| $DPN^-$ | $\mathbf{99.0_{\pm0.1}}$ | $\mathbf{99.0_{\pm0.1}}$ | $97.7_{\pm0.1}$ | $6.0_{\pm0.3}$ | $84.7_{\pm0.4}$ | $\mathbf{85.3_{\pm0.5}}$ | $84.9_{\pm0.5}$ | $34.6_{\pm0.4}$ | $99.2_{\pm0.1}$ | $\mathbf{99.3_{\pm0.0}}$ | $98.1_{\pm0.1}$ | $5.0_{\pm0.2}$ |

| OOD | Tiny [29] | | | | STL-10 [32] | | | | LSUN [33] | | | |
|---|---|---|---|---|---|---|---|---|---|---|---|---|
| | Max.P | MI | $\alpha_0$ | D.Ent | Max.P | MI | $\alpha_0$ | D.Ent | Max.P | MI | $\alpha_0$ | D.Ent |
| Baseline | $68.8_{\pm0.2}$ | - | - | - | $69.6_{\pm0.0}$ | - | - | - | $72.5_{\pm0.0}$ | - | - | - |
| MCDP | $69.7_{\pm0.3}$ | $70.6_{\pm0.3}$ | - | - | $70.7_{\pm0.1}$ | $71.6_{\pm0.2}$ | - | - | $74.5_{\pm0.1}$ | $75.9_{\pm0.2}$ | - | - |
| DE | $68.9_{\pm NA}$ | $69.6_{\pm NA}$ | - | - | $69.6_{\pm NA}$ | $70.2_{\pm NA}$ | - | - | $72.6_{\pm NA}$ | $73.4_{\pm NA}$ | - | - |
| OE | $89.5_{\pm1.0}$ | - | - | - | $91.2_{\pm0.7}$ | - | - | - | $92.2_{\pm0.9}$ | - | - | - |
| $DPN_{rev}$ | $81.2_{\pm0.2}$ | $83.8_{\pm0.1}$ | $83.8_{\pm0.1}$ | $83.5_{\pm0.1}$ | $87.2_{\pm0.1}$ | $89.3_{\pm0.1}$ | $89.3_{\pm0.1}$ | $89.0_{\pm0.1}$ | $86.7_{\pm0.0}$ | $89.3_{\pm0.1}$ | $89.3_{\pm0.1}$ | $88.9_{\pm0.1}$ |
| $DPN^+$ | $85.9_{\pm0.3}$ | $92.2_{\pm0.1}$ | $92.2_{\pm0.1}$ | $92.3_{\pm0.1}$ | $89.1_{\pm0.2}$ | $95.0_{\pm0.0}$ | $95.0_{\pm0.0}$ | $94.8_{\pm0.0}$ | $90.3_{\pm0.3}$ | $95.0_{\pm0.1}$ | $95.0_{\pm0.1}$ | $95.0_{\pm0.1}$ |
| $DPN^-$ | $89.2_{\pm0.1}$ | $\mathbf{94.5_{\pm0.1}}$ | $94.5_{\pm0.1}$ | $38.1_{\pm0.5}$ | $92.8_{\pm0.1}$ | $\mathbf{96.8_{\pm0.1}}$ | $96.8_{\pm0.1}$ | $25.4_{\pm0.4}$ | $92.8_{\pm0.1}$ | $\mathbf{96.5_{\pm0.1}}$ | $96.5_{\pm0.1}$ | $31.5_{\pm0.4}$ |

| OOD | CIFAR-10 [28] | | | | CIFAR-100 [28] | | | | Textures [34] | | | |
|---|---|---|---|---|---|---|---|---|---|---|---|---|
| | Max.P | MI | $\alpha_0$ | D.Ent | Max.P | MI | $\alpha_0$ | D.Ent | Max.P | MI | $\alpha_0$ | D.Ent |
| Baseline | $76.9_{\pm0.2}$ | - | - | - | $73.6_{\pm0.2}$ | - | - | - | $70.9_{\pm0.2}$ | - | - | - |
| MCDP | $77.4_{\pm0.1}$ | $77.5_{\pm0.2}$ | - | - | $74.0_{\pm0.2}$ | $73.6_{\pm0.2}$ | - | - | $70.3_{\pm0.2}$ | $63.6_{\pm0.2}$ | - | - |
| DE | $76.9_{\pm NA}$ | $77.7_{\pm NA}$ | - | - | $73.7_{\pm NA}$ | $75.3_{\pm NA}$ | - | - | $71.1_{\pm NA}$ | $76.2_{\pm NA}$ | - | - |
| OE | $91.3_{\pm0.4}$ | - | - | - | $89.5_{\pm0.5}$ | - | - | - | $95.8_{\pm0.3}$ | - | - | - |
| $DPN_{rev}$ | $85.4_{\pm0.7}$ | $82.8_{\pm1.4}$ | $81.9_{\pm1.6}$ | $85.6_{\pm0.9}$ | $84.2_{\pm0.8}$ | $82.5_{\pm1.4}$ | $81.7_{\pm1.6}$ | $85.0_{\pm0.9}$ | $90.9_{\pm0.3}$ | $91.2_{\pm0.6}$ | $90.6_{\pm0.6}$ | $92.6_{\pm0.3}$ |
| $DPN^+$ | $99.2_{\pm0.0}$ | $99.7_{\pm0.0}$ | $99.7_{\pm0.0}$ | $99.6_{\pm0.0}$ | $98.8_{\pm0.0}$ | $99.5_{\pm0.0}$ | $99.5_{\pm0.0}$ | $99.4_{\pm0.0}$ | $96.5_{\pm0.1}$ | $98.4_{\pm0.0}$ | $98.4_{\pm0.0}$ | $98.2_{\pm0.0}$ |
| $DPN^-$ | $99.7_{\pm0.0}$ | $\mathbf{99.9_{\pm0.0}}$ | $99.9_{\pm0.0}$ | $3.5_{\pm0.1}$ | $98.7_{\pm0.1}$ | $\mathbf{99.6_{\pm0.0}}$ | $99.6_{\pm0.0}$ | $7.5_{\pm0.2}$ | $95.8_{\pm0.1}$ | $\mathbf{98.7_{\pm0.1}}$ | $98.7_{\pm0.1}$ | $19.3_{\pm0.4}$ |

*(Left row-group labels: C10, C100, TIM for the three sub-tables respectively.)*

Tables 1 shows the performance of C10, C100 and TIM classifiers for *OOD detection* task. We observe that our $DPN^-$ models consistently outperform the other models using mutual information (MI) measure. Our $DPN^-$ models produce sharp multi-modal Dirichlet distributions for OOD examples, leading to higher MI scores compared to the in-domain examples. In contrast, $DPN^-$ models produce sharp Dirichlet distributions for both in-domain confident predictions and OOD examples. Hence, we cannot use D.Ent to distinguish them. However, in Table 3, we show that we can combine D.Ent with a total uncertainty measure to distinguish the in-domain and OOD examples.

Table 2: AUROC scores for misclassified image detection. Refer to Table 9 (Appendix) for additional results by using AUPR scores along with in-domain classification accuracy.

| | C10 | | | | C100 | | | | TIM | | | |
|---|---|---|---|---|---|---|---|---|---|---|---|---|
| | Max.P | MI | $\alpha_0$ | D.Ent | Max.P | MI | $\alpha_0$ | D.Ent | Max.P | MI | $\alpha_0$ | D.Ent |
| Baseline | $93.3_{\pm0.1}$ | - | - | - | $86.8_{\pm0.1}$ | - | - | - | $86.7_{\pm0.0}$ | - | - | - |
| MCDP | $\mathbf{93.6_{\pm0.2}}$ | $93.2_{\pm0.1}$ | - | - | $\mathbf{87.2_{\pm0.0}}$ | $83.3_{\pm0.3}$ | - | - | $86.6_{\pm0.1}$ | $83.3_{\pm0.3}$ | - | - |
| DE | $93.5_{\pm NA}$ | $92.7_{\pm NA}$ | - | - | $87.0_{\pm NA}$ | $83.4_{\pm NA}$ | - | - | $\mathbf{86.8_{\pm NA}}$ | $83.3_{\pm NA}$ | - | - |
| OE | $92.0_{\pm0.0}$ | - | - | - | $86.9_{\pm0.0}$ | - | - | - | $85.9_{\pm0.2}$ | - | - | - |
| $DPN_{rev}$ | $89.6_{\pm0.1}$ | $88.7_{\pm0.2}$ | $88.7_{\pm0.2}$ | $89.0_{\pm0.2}$ | $79.3_{\pm0.1}$ | $73.5_{\pm0.1}$ | $73.1_{\pm0.1}$ | $75.7_{\pm0.1}$ | $81.9_{\pm0.3}$ | $72.2_{\pm0.7}$ | $70.2_{\pm0.9}$ | $78.3_{\pm0.3}$ |
| $DPN^+$ | $92.2_{\pm0.3}$ | $90.3_{\pm0.1}$ | $90.3_{\pm0.1}$ | $90.5_{\pm0.2}$ | $86.5_{\pm0.1}$ | $81.2_{\pm0.0}$ | $81.3_{\pm0.0}$ | $81.9_{\pm0.1}$ | $85.7_{\pm0.2}$ | $78.3_{\pm0.4}$ | $78.7_{\pm0.5}$ | $79.7_{\pm0.2}$ |
| $DPN^-$ | $92.6_{\pm0.1}$ | $89.9_{\pm0.0}$ | $89.9_{\pm0.0}$ | $66.2_{\pm0.7}$ | $86.4_{\pm0.1}$ | $82.3_{\pm0.0}$ | $82.3_{\pm0.0}$ | $81.7_{\pm0.1}$ | $85.4_{\pm0.1}$ | $79.1_{\pm0.5}$ | $79.4_{\pm0.4}$ | $79.9_{\pm0.2}$ |

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

# Appendix

**Organization:** We organize the appendix as follows:

1. We present a set of ablation studies for our models in Section A.

2. Section B.1 provides the implementation details for our experiments on synthetic datasets.

   Section B.2 provides the experimental setup, implementation details of our models, and competitive models along with the description of the OOD test datasets for our experiments on the benchmark image classification datasets.

   Section B.3 presents a comparative study for confidence calibration performance of different models.

3. The expressions for differential entropy, mutual information of a Dirichlet distribution, and the KL Divergence between two Gaussian distributions are provided in Section C.

4. The extended results (mean $\pm$ standard deviation of 3 models) for the benchmark image classification datasets are provided in Section D from Table 9 to Table 13.

## A Ablation Studies

### A.1 Different choices for $\lambda_{in}$ and $\lambda_{out}$

Choosing both $\lambda_{in}$ and $\lambda_{out}$ to 0 lead Eqn 12 to the same loss function as non-Bayesian outlier exposure (OE) framework [15], while loosing control over the precision of the output Dirichlet distributions. In contrast, setting either $\lambda_{in}$ or $\lambda_{out}$ to 0 loses control over the precision for in-domain or OOD examples respectively. We train two additional DPN models, denoted as $\text{DPN}_{\{0,-0.5\}}$ and $\text{DPN}_{\{0.5,0\}}$ for C100 classification tasks to investigate the choices of these hyper-parameters. $\text{DPN}_{\{0,-0.5\}}$ is trained using $\lambda_{in} = 0$ and $\lambda_{out} = -0.5$. $\text{DPN}_{\{0.5,0\}}$ is trained using $\lambda_{in} = 0.5$ and $\lambda_{out} = 0$. In Table 4, we present their comparative performance with $\text{DPN}^-$.

Table 4: AUROC scores for OOD image detection of our $\text{DPN}^-$ models using different values of $\lambda_{in}$ and $\lambda_{out}$ for C100 classification task. We report (mean $\pm$ standard deviation) values of three runs.

| OOD | Tiny | | | | STL-10 | | | | LSUN | | | |
|---|---|---|---|---|---|---|---|---|---|---|---|---|
| | Max.P | MI | $\alpha_0$ | D.Ent | Max.P | MI | $\alpha_0$ | D.Ent | Max.P | MI | $\alpha_0$ | D.Ent |
| $\text{DPN}_{\{0,-0.5\}}$ | $84.6_{\pm0.0}$ | $91.1_{\pm0.1}$ | $91.6_{\pm0.2}$ | $21.7_{\pm0.7}$ | $90.2_{\pm0.1}$ | $95.0_{\pm0.0}$ | $95.7_{\pm0.1}$ | $12.1_{\pm0.3}$ | $88.5_{\pm0.2}$ | $93.8_{\pm0.1}$ | $94.2_{\pm0.1}$ | $17.5_{\pm0.3}$ |
| $\text{DPN}_{\{0.5,0\}}$ | $89.2_{\pm0.4}$ | $93.6_{\pm0.1}$ | $93.6_{\pm0.1}$ | $93.6_{\pm0.1}$ | $92.1_{\pm0.5}$ | $96.2_{\pm0.3}$ | $96.2_{\pm0.3}$ | $96.0_{\pm0.2}$ | $92.0_{\pm0.3}$ | $95.9_{\pm0.1}$ | $95.9_{\pm0.1}$ | $95.9_{\pm0.1}$ |
| $\text{DPN}^-$ | $89.2_{\pm0.1}$ | $\mathbf{94.5_{\pm0.1}}$ | $\mathbf{94.5_{\pm0.1}}$ | $38.1_{\pm0.5}$ | $92.8_{\pm0.1}$ | $\mathbf{96.8_{\pm0.1}}$ | $\mathbf{96.8_{\pm0.1}}$ | $25.4_{\pm0.4}$ | $92.8_{\pm0.1}$ | $\mathbf{96.5_{\pm0.1}}$ | $\mathbf{96.5_{\pm0.1}}$ | $31.5_{\pm0.4}$ |

**Analyzing $\text{DPN}_{\{0,-0.5\}}$:** The choice of only $\lambda_{in} = 0$ in Eqn. 12 does not enforce the DPN to produce larger concentration parameters for the in-domain examples. It only learns to produce fractional (i.e <1) concentration parameters for OOD examples, leading to produce sharper multi-modal Dirichlet distributions for OOD examples.

However, now the network can produce fractional (i.e <1) concentration parameters even for in-domain examples as well. This leads to inappropriately interpolate the concentration parameters in the boundary of in-domain and OOD regions. As a result, it leads to degrading the OOD detection performance. We can see that, similar to $\text{DPN}^-$ models, $\text{DPN}_{\{0,-0.5\}}$ models also produce lower AUROC scores for D.Ent. This indicates that the choice of $\lambda_{in} = 0, \lambda_{out} < 0$ leads to produce sharp multi-modal Dirichlet distributions for OOD examples. However, their overall OOD detection performance degrade compare to $\text{DPN}^-$ models.

**Analyzing $\text{DPN}_{\{0.5,0\}}$:** On the other hand, $\text{DPN}_{\{0.5,0\}}$ demonstrates similar property as $\text{DPN}^+$. In this case, the network produces flatter Dirichlet distributions for OOD examples compare to the in-domain examples. As we can see that $\text{DPN}_{\{0.5,0\}}$ produces high AUROC scores for D.Ent measure. However, as before, it does not address the issue of efficiently maximizing the *'representational gap'* between in-domain and OOD examples. We can see in Table 4, $\text{DPN}_{\{0.5,0\}}$ cannot exceed the OOD detection performance of $\text{DPN}^-$ models, similar to the $\text{DPN}^+$ models.

## A.2 A different choice of $\beta_{out}$ for RKL loss

In section 4, we explain that choosing fractional values for target concentration parameters, $\boldsymbol{\beta}_{out}$ for RKL loss [16] does not guarantee to produce fractional concentration parameters for OOD examples (see Eq. 9). Here, we investigate this by choosing the target concentration parameters to 0.1 for all classes for OOD training examples. For in-domain training examples, we set the target concentration parameters as 100 for the correct class and 1 for the incorrect classes. We denote it as $DPN_{rev}^{0.1}$.

In Table 5, we compare their OOD detection performance with the standard $DPN_{rev}$ models where the target concentration parameters for OOD examples are set to 1 for all classes. We observe that the performance of $DPN_{rev}^{0.1}$ models produce lower AUROC scores for D.Ent measures, while their overall performance degrade compare to the standard $DPN_{rev}$ models. This is because $DPN_{rev}^{0.1}$ models often produce both greater than and less than 1 values of concentration parameters of OOD examples, that leads to uni-modal Dirichlet distributions, instead of a multi-modal Dirichlet. This representation is often similar to the in-domain examples. Hence, it becomes even more difficult to distinguish the in-domain and OOD examples for $DPN_{rev}^{0.1}$ models, which lead to degrade their overall performance.

Table 5: AUROC scores OOD image detection results for DPN models using RKL loss function [16] with different choices of hyper-parameters for C100 classification task. We report (mean $\pm$ standard deviation) values of three runs.

| OOD | Tiny | | | | STL-10 | | | | LSUN | | | |
|---|---|---|---|---|---|---|---|---|---|---|---|---|
| | Max.P | MI | $\alpha_0$ | D.Ent | Max.P | MI | $\alpha_0$ | D.Ent | Max.P | MI | $\alpha_0$ | D.Ent |
| $DPN_{rev}^{0.1}$ | $74.9_{\pm0.2}$ | $80.4_{\pm0.2}$ | $80.7_{\pm0.2}$ | $48.1_{\pm0.1}$ | $78.1_{\pm0.1}$ | $84.3_{\pm0.1}$ | $84.7_{\pm0.1}$ | $32.8_{\pm0.2}$ | $76.7_{\pm0.1}$ | $82.2_{\pm0.0}$ | $82.5_{\pm0.1}$ | $49.3_{\pm0.2}$ |
| $DPN_{rev}$ | $81.2_{\pm0.2}$ | $\mathbf{83.8_{\pm0.1}}$ | $\mathbf{83.8_{\pm0.1}}$ | $83.5_{\pm0.1}$ | $87.2_{\pm0.1}$ | $\mathbf{89.3_{\pm0.1}}$ | $\mathbf{89.3_{\pm0.1}}$ | $89.0_{\pm0.1}$ | $86.7_{\pm0.0}$ | $\mathbf{89.3_{\pm0.1}}$ | $\mathbf{89.3_{\pm0.1}}$ | $88.9_{\pm0.1}$ |

## A.3 A Binary Classifier for OOD Detection

In this work, we show that in the presence of high data uncertainty, the existing OOD detectors often lead to the same representation for in-domain examples as the OOD examples. Hence, one can simply think of training a binary classifier using in-domain and OOD training examples as two different classes to distinguish between in-domain examples and OOD examples. Since it does not need to classify the in-domain examples among multiple classes, it would not suffer from data uncertainty and should automatically solve the problem. However, note that such a binary classifier only learns to produce sharp categorical distributions for these training examples. Hence, given an unknown OOD test example, it does not necessarily produce sharp categorical distribution for the OOD class.

For example, for our experiment on C10 classification task, we use CIFAR-10 training images as the in-domain training set and CIFAR-100 training examples the OOD training set. In contrast, for C100 classification task, we use CIFAR-100 training images as the in-domain training set and CIFAR-10 training examples as the OOD training set. Hence, given an OOD test example from TIM dataset, if the binary classifier for C10 produces a higher probability score for the OOD class, it is expected to produce a lower probability score for the OOD class for C100 classification task. In contrast, our $DPN^-$ models are explicitly trained to produce multi-modal Dirichlet distributions for an unknown test example, whenever it does not 'fit' into the in-domain class-labels.

Table 6: OOD image detection results of the binary classifiers compare to our $DPN^-$ models for C10 and C100 classification task. We report (mean $\pm$ standard deviation) values of three runs.

| OOD | C-10 Classification | | | | C-100 classification | | | |
|---|---|---|---|---|---|---|---|---|
| | AUROC | | AUPR | | AUROC | | AUPR | |
| | Binary | $DPN^-$ | Binary | $DPN^-$ | Binary | $DPN^-$ | Binary | $DPN^-$ |
| Tiny | $\mathbf{99.0_{\pm0.2}}$ | $\mathbf{99.0_{\pm0.1}}$ | $\mathbf{99.1_{\pm0.1}}$ | $\mathbf{99.1_{\pm0.1}}$ | $84.8_{\pm1.2}$ | $\mathbf{94.5_{\pm0.1}}$ | $87.9_{\pm0.7}$ | $\mathbf{95.1_{\pm0.1}}$ |
| STL-10 | $\mathbf{93.7_{\pm0.9}}$ | $85.3_{\pm0.5}$ | $\mathbf{94.0_{\pm0.7}}$ | $85.2_{\pm0.6}$ | $90.7_{\pm0.9}$ | $\mathbf{96.8_{\pm0.1}}$ | $91.0_{\pm0.6}$ | $\mathbf{96.7_{\pm0.1}}$ |
| LSUN | $98.9_{\pm0.2}$ | $\mathbf{99.3_{\pm0.0}}$ | $98.9_{\pm0.2}$ | $\mathbf{99.3_{\pm0.1}}$ | $91.2_{\pm1.2}$ | $\mathbf{96.5_{\pm0.1}}$ | $93.0_{\pm0.8}$ | $\mathbf{97.0_{\pm0.1}}$ |
| Places365 | $98.8_{\pm0.2}$ | $\mathbf{98.9_{\pm0.1}}$ | $\mathbf{99.7_{\pm0.0}}$ | $\mathbf{99.7_{\pm0.0}}$ | $88.2_{\pm1.3}$ | $\mathbf{94.5_{\pm0.1}}$ | $96.7_{\pm0.3}$ | $\mathbf{98.4_{\pm0.0}}$ |
| Textures | $\mathbf{99.7_{\pm0.1}}$ | $\mathbf{99.7_{\pm0.0}}$ | $\mathbf{99.6_{\pm0.1}}$ | $99.4_{\pm0.1}$ | $65.4_{\pm1.5}$ | $\mathbf{85.2_{\pm0.1}}$ | $65.2_{\pm0.9}$ | $\mathbf{78.9_{\pm0.2}}$ |

In Table 6, we compare the OOD detection performance of such binary classifiers with our $DPN^-$ models for C10 and C100. For the binary classifier, we consider the probability score of the "in-domain" class as their uncertainty metric. For our $DPN^-$ models, we select the best AUROC and

AUPR values from Table 11 and Table 12 for C10 and C100 classification tasks respectively.. We can see that, while the binary classifier often out-performs our DPN$^-$ models, it does not necessarily provide the upper bound for the OOD detection tasks. In practice, since we do not know the characteristics of the OOD test examples, it may not be suitable to use a binary classifier for OOD detection tasks.

Figure 7: Visualization and understanding the desired characteristics of different DPN models. We visualize the uncertainty measures for different data-points for DPN$_{rev}$, DPN$^+$ and DPN$^-$.

(a) In-domain & OOD training examples

(b) DPN$_{rev}$

(c) DPN$^+$

(d) DPN$^-$

# B    Implementation Details and Extended Results

## B.1    Synthetic Datasets

The three classes of our synthetic dataset are constructed by sampling from three different isotropic Gaussian distributions with means of $(-4, 0)$, $(4, 0)$ and $(0, 5)$ and isotropic variances of $\sigma = 4$. We sample 200 training data points from each distribution for each class. We also draw 600 samples for OOD training examples from a uniform distribution of $\mathcal{U}([-15, 15], [-13, 17])$, outside the Gaussian distributions.

**Hyper-parameters.** We train a neural network with 2 hidden layers with 125 nodes each and $relu$ activation function. The training code is provided along with the supplementary materials.

We use a neural network with 2 hidden layers with 50 nodes each and ReLU activation function. We set $\gamma = 1.0$ in the overall loss function. We have two DPN models. Our first DPN model, DPN$^+$, is trained using a positive $\lambda_{out} = \frac{1}{\#class} + 0.5$ and $\lambda_{in} = 1.5$. The second DPN model, DPN$^-$, is trained using a negative $\lambda_{out} = \frac{1}{\#class} - 0.5$, and $\lambda_{in} = 0.5$. Our DPN$^+$ and DPN$^-$ models are trained using the SGD optimizer.

We also train a DPN$_{rev}$ model by using RKL loss [16]. We set the concentration hyper-parameters as follows: for in-domain training examples, we set the concentration parameters to $1e2$ for the correct class and 1 for the incorrect classes. For the OOD training examples, we set the concentration parameters as 1 for all classes. We train the DPN$_{rev}$ model using ADAM optimizer [37]. Notably, we could not train the DPN$_{rev}$ model using SGD optimization due to their complex RKL loss function.

**Additional Results.** We visualize the uncertainty measures of different data points for DPN$_{rev}$ along with our DPN$^+$ and DPN$^-$ models in Figure 7(b), 7(c) and 7(d) respectively. We observe very similar characteristics for DPN$_{rev}$ and our DPN$^+$. In contrast, our DPN$^-$ produces much sharper boundaries to distinguish the in-domain and OOD examples using distributional uncertainty measure i.e mutual information and precision (or inverse-EPKL) to demonstrate its superiority.

We also present the results for $entropy$ of categorical posterior distributions, $\mathcal{H}[p(\omega_c|\boldsymbol{x}^*, D)]$, for different data points. This is a total uncertainty measure as it is derived from the expected predictive categorical distribution, $p(\omega_c|\boldsymbol{x}^*, D)$ i.e by marginalizing $\boldsymbol{\mu}$ and $\boldsymbol{\theta}$ in Eq. 2.

## B.2 Benchmark Image Classification Datasets

We use the VGG-16 network for C10, C100, and TIM classification tasks. For C10, we use CIFAR-10 training images ($50,000$ images) as our in-domain training data and CIFAR-100 training images ($50,000$ images) as our OOD training data.

For C-100, we use CIFAR-100 training images ($50,000$ images) as our in-domain training data and CIFAR-10 training images ($50,000$ images) as our OOD training data.

For TIM, we use TIM training images (100,000 images) as our in-domain training data and ImageNet-25K images ($25,000$ images) as our OOD training data. ImageNet-25K is obtained by randomly selecting $25,000$ images from the ImageNet dataset [30].

**Hyper-parameters for our DPN$^+$ & DPN$^-$ models.** Similar to [5, 16, 15], we do not need to tune any hyper-parameters during testing. In other words, the OOD test examples remain unknown to our DPN classifiers, as in a real-world scenario. We set $\gamma = 0.5$ for our loss function in Eqn. 12, as applied in [15].

We train two different DPN models for each classification task, using both positive and negative values for $\lambda_{out}$ to analyze the effect of flat Dirichlet distributions and sharp Dirichlet distributions across the edges of the simplex respectively for the OOD examples. Note that we cannot choose arbitrarily large values for $\lambda_{in}$ and $\lambda_{out}$ as it would degrade the in-domain classification accuracy of the models. For our experiments in the main paper, we select the hyper-parameters as follows: Our first model, DPN$^+$ is trained with positive $\lambda_{out} = \frac{1}{\#class} + 0.5$ and $\lambda_{in} = 1.5$. The second model, DPN$^-$ is trained with negative $\lambda_{out} = \frac{1}{\#class} - 0.5$, and $\lambda_{in} = 0.5$.

**Competitive Systems: Implementations and Additional Discussions.** We compare the performance of our models with standard DNN as baseline model [31], Bayesian Monte-Carlo dropout (MCDP) [11], deep-ensemble models (DE) [13] and evidential deep learning (EDL) [18], DPN$_{fwd}$ and DPN$_{rev}$ using the forward and reverse KL-divergence loss function proposed in [5] and [16] respectively, non-Bayesian frameworks such as outlier exposure (OE) [15]. We use the same architecture as our DPN models for the other competitive models.

**MCDP models:** For MCDP models, we use the standard DNN (baseline) model with randomly dropping the nodes during test time. The predictive categorical distributions are obtained by averaging the outputs for 10 iterations. For OE, DPN$_{fwd}$ and DPN$_{rev}$ models, we use the same setup as applied

Table 7: Details of Train and Test Datasets used for the different classifiers.

| Classifier | Input | #Classes | Training Datasets | | Test Datasets | |
|---|---|---|---|---|---|---|
| | | | In-Domain | OOD | In-Domain | OOD |
| C10 | $32 \times 32$ | 10 | CIFAR-10 Training Set (50,000 images) | CIFAR-100 Training Set (50,000 images) | CIFAR-10 Test Images (10,000 Images) | Tiny, STL-10, LSUN etc. |
| C100 | $32 \times 32$ | 100 | CIFAR-100 Training Set (50,000 images) | CIFAR-10 Training Set (50,000 images) | CIFAR-100 Test Set (10,000 Images) | Tiny, STL-10, LSUN etc. |
| TIM | $64 \times 64$ | 200 | TinyImageNet Training Set (100,000 images) | ImageNet-25K (25,000 randomly sampled images from ImageNet) | TIM Test Images (10,000 Images) | CIFAR-10, CIFAR-100, Textures etc. |

for our DPN models (See Table 7). Table 9 presents the classification accuracies achieved by different models for different classification tasks, along with their performance for misclassified example detection. For DE, we use an ensemble of three baseline DNN models for our experiments.

**EDL models:** Similar to DPN models, EDL also produces Dirichlet distribution for the input examples [18]. Unlike DPN, EDL applies ReLU activation, instead of the exponential function (Eqn. 5), to induce non-negative constraints to produce the concentration parameters of their output Dirichlet distributions.

EDL models are trained using the in-domain examples. Their network is trained using a loss function that explicitly maximizes the concentration parameter of the correct class while minimizing the overall precision of the Dirichlet for each in-domain training examples. For C10 classifiers, we train the EDL models from the scratch. For C100 and TIM classifiers, we initialize the EDL models using the pre-trained Baseline models to achieve competitive in-domain classification accuracy. Next, we replace the soft-max using ReLU activation and retrain the models using the proposed loss function for EDL.

Since both DPN and EDL models output Dirichlet distribution for an input example, we can define the same uncertainty measures as the DPN models. In Table 9 and Table 11-13, we present the results of the EDL models. Interestingly, in Table 9, we observe that EDL models often tend to produce lower AUROC and AUPR scores under distributional uncertainty measures for the misclassification detection task. This property is desirable (see Section 4 and 5.1). However, they achieve significantly lower OOD detection performance compared to the state-of-the-art competitive models (Table 11-13). Further, in Table 8, we observe that the calibration performance of EDL models are also dropped than the other state-of-the-art OOD detection models.

**Existing DPN models [5, 16]:** $DPN_{fwd}$ and $DPN_{rev}$ models are trained only using the ADAM optimizer for all classification tasks [37]. We could not use the SGD optimizer to train these models due to the complex RKL loss. In contrast, we have not encountered such a problem for other models.

For example, we use SGD to train the other models for C10 classification task. We observe that both $DPN_{fwd}$ and $DPN_{rev}$ models achieve lower classification accuracy than the other classifiers for this task (Table 9). For C100 and TIM classification tasks, we choose the ADAM optimizer for all models. We find that all the OOD detection models achieve similar classification accuracy for these two tasks (Table 9).

Further, note that, the choice of larger values for the hyper-parameter, $\beta$ for the RKL in Eqn. 8 makes it difficult to optimize the network. For our experiments, we choose the same set of hyper-parameters for $DPN_{fwd}$ and $DPN_{rev}$ models as suggested in the original paper [16]. The concentration parameters for in-domain training examples are set to 100 for the correct class and 1 for the incorrect classes. For OOD training examples, we choose the concentration parameters as 1 for all classes.

### B.2.1 Description of the OOD Test Datasets

We use a wide-range of OOD datasets for our experiments, as described in the following. For C10 and C100 classifiers, these input test images are resized to $32 \times 32$, while for TIM classifiers, we resize them to $64 \times 64$.

*TinyImageNet (Tiny)* [29]. This dataset is used as an OOD test dataset *only* for C-10 and C-100 classifiers. Note that, for TinyImageNet classifiers, this is the in-domain test set.

This is a subset of the Imagenet dataset. We use the validation set, that contains $10,000$ test images from 200 different image classes for our evaluation during test time.

*CIFAR-10 and CIFAR-100* [28]. This dataset is used as the OOD test dataset *only* for TIM classifiers. We use the validation set, that contains $10,000$ test images from 10 and 100 image classes respectively.

*LSUN* [33]. The Large-scale Scene UNderstanding dataset (LSUN) contains images of 10 different scene categories. We use its validation set, containing $10,000$ images, as an unknown OOD test set.

*Places 365* [38]. The validation set of this dataset consists of 36500 images of 365 scene categories.

*Textures* [34] contains 5640 textural images in the wild belonging to 47 categories.

*STL-10* contains $8,000$ images of natural images from 10 different classes [32].

## B.3 Results for Confidence Calibration

Calibration error measures if the confidence estimates produced by the classifier for its predictions misrepresent the empirical performance [39, 35, 15]. A well-calibrated classifier should produce the confidence probabilities that matches with the empirical frequency of correctness. For example, if a classifier predicts an event with 90% probability, we would like it to be corrected for 90% of the time. However, several studies have demonstrated that DNN classifiers tend to produce over-confidence in their predictions.

Several measures have been proposed to compute the calibration error of a classifier [35, 15]. In this paper, we use the Root Mean Square (RMS) Calibration Error that computes the square root of the expected squared difference between confidence and accuracy at a confidence level [15]. Since the confidence values can be distributed non-uniformly, Hendrycks et al. [15] proposed to partition the samples into multiple bins with dynamic ranges and measure the average confidence and accuracy of each bin to compute the calibration error.

A real-world classifier should provide calibrated probabilities on both in- and out-of-distribution examples. Hence, Hendrycks et al. [15] proposed to incorporate OOD test examples in the calculation of the RMS calibration error. Since the OOD examples do not belong to any of the in-domain classes, these examples are always considered to be incorrectly classified. Hence, the classifier should produce low confidence for these OOD test examples.

Table 8: Root mean square (RMS) calibration error. Lower scores are better.

| | C10 | C100 | TIM |
|---|---|---|---|
| Baseline | $16.2_{\pm 0.0}$ | $6.6_{\pm 0.3}$ | $5.2_{\pm 0.0}$ |
| MCDP | $15.7_{\pm 0.1}$ | $6.7_{\pm 0.0}$ | $5.3_{\pm 0.2}$ |
| DE | $16.1_{\pm NA}$ | $6.8_{\pm NA}$ | $6.2_{\pm NA}$ |
| EDL | $14.9_{\pm 0.2}$ | $17.7_{\pm 0.6}$ | $12.4_{\pm 0.1}$ |
| OE | $6.4_{\pm 0.4}$ | $3.8_{\pm 0.1}$ | $4.2_{\pm 0.1}$ |
| $DPN_{rev}$ | $9.2_{\pm 0.4}$ | $10.4_{\pm 0.1}$ | $7.2_{\pm 0.5}$ |
| $DPN^+$ | $\mathbf{6.3_{\pm 0.3}}$ | $4.3_{\pm 0.0}$ | $2.8_{\pm 0.3}$ |
| $DPN^-$ | $6.5_{\pm 0.2}$ | $\mathbf{3.5_{\pm 0.1}}$ | $\mathbf{2.7_{\pm 0.3}}$ |

In Table 8 we present a comparative results for RMS calibration error [15]. We take an equal number of OOD test examples as the in-domain test samples for this experiment. For C10 and C100 classifiers, we use $5,000$ in-domain test examples from and take $5,000$ OOD test examples from STL-10 dataset. For TIM classifiers, we use $5,000$ in-domain test examples and $5,000$ OOD test examples from CIFAR-100. We apply the soft-max temperature scaling to report the calibration error in Table 8. Note that, EDL models do not apply soft-max activation to produce their probability scores. Hence, we use 'temperature translating' where we instead add the temperature parameter to the logit outputs. We can Table 8 that our proposed $DPN^-$ models achieve comparable performance with the OE and $DPN^+$ models for C10. For C100 and TIM, our $DPN^-$ models outperform other comparative systems.

## C Derivations of different measures

### C.1 Differential Entropy for a Dirichlet

Differential Entropy of a Dirichlet distribution can be calculated as follows:

$$\mathcal{H}[p(\boldsymbol{\mu}|\boldsymbol{x}^*, D_{in})] = -\int p(\boldsymbol{\mu}|\boldsymbol{x}^*, D_{in}) \ln p(\boldsymbol{\mu}|\boldsymbol{x}^*, D_{in}) d\boldsymbol{\mu}$$

$$= \sum_{c=1}^{K} \ln \Gamma(\alpha_c) - \ln \Gamma(\alpha_0) - \sum_{c=1}^{K} (\alpha_c - 1)(\psi(\alpha_c) - \psi(\alpha_0)) \quad (14)$$

where, $\alpha_c$ is a function of $x^*$. $\Gamma$ and $\psi$ denotes the Gamma and digamma functions respectively.

## C.2  Mutual Information of a Dirichlet

The mutual information of the labels $y$ and the categorical $\boldsymbol{\mu}$ of a DPN is computed as:

$$\mathcal{I}[y, \boldsymbol{\mu}|\boldsymbol{x}^*, \hat{\boldsymbol{\theta}}] = \sum_{c=1}^{K} \frac{\alpha_c}{\alpha_0} \big[ \psi(\alpha_c + 1) - \psi(\alpha_0 + 1) - \ln \frac{\alpha_c}{\alpha_0} \big] \tag{15}$$

## C.3  KL Divergence between two Gaussians

The KL divergence from a Gaussian distribution $\mathcal{N}_1(\boldsymbol{\mu}_1, \Sigma_1)$ to Gaussian distribution, $\mathcal{N}_2(\boldsymbol{\mu}_2, \Sigma_2)$ is computed as follows:

$$KL(\, \mathcal{N}_2 \,\|\, \mathcal{N}_1) := \frac{1}{2}\Big[ tr(\Sigma_1^{-1}\Sigma_2) \; - d + \frac{det(\Sigma_1)}{det(\Sigma_0)} + (\boldsymbol{\mu_1} - \boldsymbol{\mu_2})^T \Sigma_1^{-1}(\boldsymbol{\mu_1} - \boldsymbol{\mu_2}) \Big] \tag{16}$$

where, $d$ is the dimension of $\boldsymbol{\mu}_1$ or $\boldsymbol{\mu}_2$. $det(\Sigma)$ represents the determinant of $\Sigma$. $tr$ computes the trace of a matrix.

# D   Extended Results

In the following, we present an extended version of the results for a wide range of OOD test datasets along with an additional uncertainty measure i.e, entropy. We also report the results for the deep ensemble (DE) framework using an ensemble of 3 models.

In Table 9, we present the results for misclassification detection along with the classification accuracy for different approaches. Note that the DPN models achieve higher AUROC and AUPR scores even for distributional uncertainty measures such as MI, precision ($\alpha_0$), and D.Ent.

In Table 10, we present an extended version of Table 3 along with additional OOD datasets for each classification tasks. In Table 11-13, we present the additional results for OOD detection performance.

Table 9: Classification accuracy and misclassified image detection. Here, we report the (mean ± standard deviation) of 3 runs for each framework. Note that AUPR may *not* be an ideal metric for comparison, as it depends on the *base rates* i.e the number of misclassified examples v.s correctly classified predictions. That is, AUPR scores are comparable when the models achieve similar classification accuracy. Our DPN$^-$ models achieve comparable performance foe misclassified image detection using the AUROC metric.

(a) C10 classification task

| | AUROC | | | | | AUPR | | | | | Acc. |
|---|---|---|---|---|---|---|---|---|---|---|---|
| | Max.P | Ent | MI | $\alpha_0$ | D.Ent | Max.P | Ent | MI | $\alpha_0$ | D.Ent | |
| Baseline | 93.3$_{\pm0.1}$ | 93.4$_{\pm0.1}$ | - | - | - | 43.9$_{\pm0.7}$ | 47.0$_{\pm0.3}$ | - | - | - | 94.1$_{\pm0.0}$ |
| MCDP | **93.6**$_{\pm0.2}$ | **93.6**$_{\pm0.2}$ | 93.2$_{\pm0.1}$ | - | - | 46.1$_{\pm2.0}$ | 46.5$_{\pm1.9}$ | 40.9$_{\pm1.5}$ | - | - | 94.2$_{\pm0.1}$ |
| DE | 93.5$_{\pm\text{NA}}$ | 93.5$_{\pm\text{NA}}$ | 92.7$_{\pm\text{NA}}$ | - | - | 45.6$_{\pm\text{NA}}$ | 46.6$_{\pm\text{NA}}$ | 39.8$_{\pm\text{NA}}$ | - | - | 94.0$_{\pm\text{NA}}$ |
| EDL | 91.3$_{\pm0.0}$ | 91.2$_{\pm0.0}$ | 80.1$_{\pm0.5}$ | 76.7$_{\pm0.7}$ | 89.7$_{\pm0.0}$ | 44.8$_{\pm0.2}$ | 43.8$_{\pm0.1}$ | 21.7$_{\pm0.5}$ | 18.1$_{\pm0.5}$ | 37.9$_{\pm0.1}$ | 93.1$_{\pm0.0}$ |
| OE | 92.0$_{\pm0.0}$ | 91.6$_{\pm0.0}$ | - | - | - | 35.3$_{\pm0.8}$ | 33.6$_{\pm0.8}$ | - | - | - | 94.2$_{\pm0.1}$ |
| DPN$_{fwd}$ | 90.3$_{\pm0.2}$ | 90.1$_{\pm0.2}$ | 88.6$_{\pm0.2}$ | 88.0$_{\pm0.2}$ | 88.0$_{\pm0.1}$ | 49.2$_{\pm0.9}$ | 47.7$_{\pm0.8}$ | 43.3$_{\pm0.4}$ | 41.6$_{\pm0.4}$ | 40.8$_{\pm0.6}$ | 88.3$_{\pm0.2}$ |
| DPN$_{rev}$ | 89.6$_{\pm0.1}$ | 89.4$_{\pm0.1}$ | 88.7$_{\pm0.2}$ | 88.7$_{\pm0.2}$ | 89.0$_{\pm0.2}$ | **50.0**$_{\pm0.8}$ | 48.8$_{\pm0.7}$ | 46.1$_{\pm0.9}$ | 45.8$_{\pm0.8}$ | 47.7$_{\pm0.7}$ | 90.6$_{\pm0.0}$ |
| DPN$^+$ | 92.2$_{\pm0.3}$ | 91.7$_{\pm0.3}$ | 90.3$_{\pm0.1}$ | 90.3$_{\pm0.1}$ | 90.5$_{\pm0.2}$ | 36.6$_{\pm0.5}$ | 34.9$_{\pm0.7}$ | 31.2$_{\pm0.8}$ | 31.2$_{\pm0.6}$ | 31.6$_{\pm0.7}$ | 94.0$_{\pm0.1}$ |
| DPN$^-$ | 92.6$_{\pm0.1}$ | 92.2$_{\pm0.1}$ | 89.9$_{\pm0.0}$ | 89.9$_{\pm0.0}$ | 66.2$_{\pm0.7}$ | 37.2$_{\pm0.7}$ | 35.1$_{\pm0.6}$ | 31.3$_{\pm0.4}$ | 30.6$_{\pm0.4}$ | 17.1$_{\pm0.4}$ | **94.4**$_{\pm0.0}$ |

(b) C100 classification task

| | AUROC | | | | | AUPR | | | | | Acc. |
|---|---|---|---|---|---|---|---|---|---|---|---|
| | Max.P | Ent | MI | $\alpha_0$ | D.Ent | Max.P | Ent | MI | $\alpha_0$ | D.Ent | |
| Baseline | 86.8$_{\pm0.1}$ | 87.0$_{\pm0.1}$ | - | - | - | 68.4$_{\pm0.4}$ | 69.2$_{\pm0.3}$ | - | - | - | 72.3$_{\pm0.0}$ |
| MCDP | 87.2$_{\pm0.0}$ | **87.3**$_{\pm0.0}$ | 83.3$_{\pm0.3}$ | - | - | 69.1$_{\pm0.3}$ | 69.3$_{\pm0.3}$ | 53.9$_{\pm0.5}$ | - | - | **72.7**$_{\pm0.1}$ |
| DE | 87.0$_{\pm\text{NA}}$ | 87.1$_{\pm\text{NA}}$ | 83.4$_{\pm\text{NA}}$ | - | - | 69.2$_{\pm\text{NA}}$ | **69.7**$_{\pm\text{NA}}$ | 56.2$_{\pm\text{NA}}$ | - | - | 72.2$_{\pm\text{NA}}$ |
| EDL | 85.8$_{\pm0.3}$ | 85.0$_{\pm0.3}$ | 44.4$_{\pm1.0}$ | 43.4$_{\pm1.1}$ | 55.7$_{\pm0.7}$ | 69.3$_{\pm1.1}$ | 68.5$_{\pm1.0}$ | 28.5$_{\pm1.0}$ | 28.0$_{\pm1.0}$ | 36.3$_{\pm1.4}$ | 70.4$_{\pm0.3}$ |
| OE | 86.9$_{\pm0.0}$ | 86.9$_{\pm0.1}$ | - | - | - | 67.7$_{\pm0.3}$ | 66.9$_{\pm0.4}$ | - | - | - | 71.6$_{\pm0.0}$ |
| DPN$_{rev}$ | 79.3$_{\pm0.1}$ | 78.5$_{\pm0.1}$ | 73.5$_{\pm0.1}$ | 73.1$_{\pm0.1}$ | 75.7$_{\pm0.1}$ | 65.3$_{\pm0.4}$ | 64.1$_{\pm0.3}$ | 58.4$_{\pm0.3}$ | 57.9$_{\pm0.3}$ | 61.2$_{\pm0.3}$ | 71.1$_{\pm0.1}$ |
| DPN$^+$ | 86.5$_{\pm0.1}$ | 86.5$_{\pm0.1}$ | 81.2$_{\pm0.0}$ | 81.3$_{\pm0.0}$ | 81.9$_{\pm0.1}$ | 66.8$_{\pm0.3}$ | 66.3$_{\pm0.3}$ | 57.8$_{\pm0.2}$ | 57.8$_{\pm0.2}$ | 59.2$_{\pm0.3}$ | 72.1$_{\pm0.1}$ |
| DPN$^-$ | 86.4$_{\pm0.1}$ | 86.5$_{\pm0.1}$ | 82.3$_{\pm0.0}$ | 82.3$_{\pm0.0}$ | 81.7$_{\pm0.1}$ | 67.0$_{\pm0.5}$ | 66.6$_{\pm0.3}$ | 58.9$_{\pm0.2}$ | 58.9$_{\pm0.2}$ | 59.1$_{\pm0.2}$ | 72.3$_{\pm0.1}$ |

(c) TIM classification task

| | AUROC | | | | | AUPR | | | | | Acc. |
|---|---|---|---|---|---|---|---|---|---|---|---|
| | Max.P | Ent | MI | $\alpha_0$ | D.Ent | Max.P | Ent | MI | $\alpha_0$ | D.Ent | |
| Baseline | 86.7$_{\pm0.0}$ | 86.8$_{\pm0.1}$ | - | - | - | 77.2$_{\pm0.1}$ | 77.1$_{\pm0.3}$ | - | - | - | 62.5$_{\pm0.2}$ |
| MCDP | 86.6$_{\pm0.1}$ | 86.4$_{\pm0.1}$ | 83.3$_{\pm0.3}$ | - | - | 76.8$_{\pm0.3}$ | 76.4$_{\pm0.3}$ | 67.2$_{\pm1.2}$ | - | - | **62.7**$_{\pm0.2}$ |
| DE | **86.8**$_{\pm\text{NA}}$ | **86.8**$_{\pm\text{NA}}$ | 83.3$_{\pm\text{NA}}$ | - | - | 77.2$_{\pm\text{NA}}$ | 77.0$_{\pm\text{NA}}$ | 67.6$_{\pm\text{NA}}$ | - | - | 62.6$_{\pm\text{NA}}$ |
| EDL | 85.9$_{\pm0.2}$ | 83.6$_{\pm0.1}$ | 73.0$_{\pm0.6}$ | 72.7$_{\pm0.6}$ | 75.5$_{\pm0.4}$ | 77.0$_{\pm0.4}$ | 73.2$_{\pm0.6}$ | 62.1$_{\pm0.5}$ | 61.9$_{\pm0.6}$ | 64.5$_{\pm0.4}$ | 60.9$_{\pm0.1}$ |
| OE | 85.9$_{\pm0.2}$ | 85.8$_{\pm0.1}$ | - | - | - | **77.7**$_{\pm0.4}$ | 77.3$_{\pm0.2}$ | - | - | - | 59.8$_{\pm0.2}$ |
| DPN$_{rev}$ | 81.9$_{\pm0.3}$ | 81.0$_{\pm0.2}$ | 72.2$_{\pm0.7}$ | 70.2$_{\pm0.9}$ | 78.3$_{\pm0.3}$ | 75.0$_{\pm0.3}$ | 73.4$_{\pm0.4}$ | 61.4$_{\pm0.9}$ | 59.2$_{\pm0.9}$ | 70.0$_{\pm0.5}$ | 60.5$_{\pm0.2}$ |
| DPN$^+$ | 85.7$_{\pm0.2}$ | 85.7$_{\pm0.1}$ | 78.3$_{\pm0.4}$ | 78.7$_{\pm0.5}$ | 79.7$_{\pm0.2}$ | 77.4$_{\pm0.4}$ | 76.7$_{\pm0.2}$ | 66.3$_{\pm0.5}$ | 66.4$_{\pm0.5}$ | 68.5$_{\pm0.4}$ | 59.7$_{\pm0.1}$ |
| DPN$^-$ | 85.4$_{\pm0.1}$ | 85.0$_{\pm0.0}$ | 79.1$_{\pm0.5}$ | 79.4$_{\pm0.4}$ | 79.9$_{\pm0.2}$ | 76.9$_{\pm0.1}$ | 76.2$_{\pm0.1}$ | 67.3$_{\pm0.7}$ | 67.4$_{\pm0.7}$ | 69.4$_{\pm0.4}$ | 59.4$_{\pm0.1}$ |

Table 10: KL-divergence scores from the distribution of uncertainty values of missclassified and correctly predicted examples to the OOD examples. Higher scores are desirable as it indicates greater gap between in-domain and OOD examples. We report the (mean $\pm$ standard deviation) of 3 runs for each frameworks.

(a) C10 classification task

| OOD | STL-10 | | Tiny | | LSUN | | Places365 | | Textures | |
|---|---|---|---|---|---|---|---|---|---|---|
| | Miss | Correct | Miss | Correct | Miss | Correct | Miss | Correct | Miss | Correct |
| $DPN_{fwd}$ | $\mathbf{1.6}_{\pm 0.1}$ | $1.6_{\pm 0.4}$ | $1.1_{\pm 0.2}$ | $6.1_{\pm 0.8}$ | $1.4_{\pm 0.2}$ | $6.7_{\pm 0.7}$ | $1.3_{\pm 0.2}$ | $6.6_{\pm 0.7}$ | $2.8_{\pm 0.4}$ | $12.7_{\pm 3.0}$ |
| $DPN_{rev}$ | $0.1_{\pm 0.0}$ | $5.7_{\pm 0.7}$ | $1.4_{\pm 0.2}$ | $12.6_{\pm 0.8}$ | $2.1_{\pm 0.3}$ | $13.9_{\pm 1.0}$ | $1.6_{\pm 0.2}$ | $13.2_{\pm 0.8}$ | $3.5_{\pm 0.1}$ | $16.3_{\pm 0.6}$ |
| $DPN^{+}$ | $0.3_{\pm 0.0}$ | $4.7_{\pm 0.5}$ | $1.5_{\pm 0.2}$ | $12.1_{\pm 1.9}$ | $2.0_{\pm 0.2}$ | $12.7_{\pm 2.0}$ | $1.5_{\pm 0.2}$ | $12.2_{\pm 1.9}$ | $\mathbf{3.7}_{\pm 0.1}$ | $15.4_{\pm 1.9}$ |
| $DPN^{-}$ | $0.5_{\pm 0.0}$ | $\mathbf{10.6}_{\pm 0.8}$ | $\mathbf{2.1}_{\pm 0.1}$ | $\mathbf{20.7}_{\pm 1.2}$ | $\mathbf{2.4}_{\pm 0.1}$ | $\mathbf{22.5}_{\pm 1.3}$ | $\mathbf{2.1}_{\pm 0.1}$ | $\mathbf{20.9}_{\pm 1.2}$ | $2.9_{\pm 0.1}$ | $\mathbf{20.4}_{\pm 1.2}$ |

(a) C100 classification task

| OOD | STL-10 | | Tiny | | LSUN | | Places365 | | Textures | |
|---|---|---|---|---|---|---|---|---|---|---|
| | Miss | Correct | Miss | Correct | Miss | Correct | Miss | Correct | Miss | Correct |
| $DPN_{rev}$ | $2.8_{\pm 0.1}$ | $9.4_{\pm 0.1}$ | $1.8_{\pm 0.0}$ | $6.6_{\pm 0.1}$ | $2.6_{\pm 0.0}$ | $8.6_{\pm 0.1}$ | $2.1_{\pm 0.0}$ | $7.3_{\pm 0.1}$ | $1.1_{\pm 0.0}$ | $4.0_{\pm 0.1}$ |
| $DPN^{+}$ | $2.7_{\pm 0.0}$ | $9.3_{\pm 0.3}$ | $1.8_{\pm 0.0}$ | $7.2_{\pm 0.2}$ | $2.6_{\pm 0.0}$ | $9.1_{\pm 0.2}$ | $2.1_{\pm 0.0}$ | $7.8_{\pm 0.2}$ | $0.5_{\pm 0.0}$ | $3.5_{\pm 0.1}$ |
| $DPN^{-}$ | $\mathbf{61.2}_{\pm 3.6}$ | $\mathbf{330.1}_{\pm 34.2}$ | $\mathbf{44.5}_{\pm 2.8}$ | $\mathbf{244.2}_{\pm 25.2}$ | $\mathbf{49.9}_{\pm 3.5}$ | $\mathbf{272.0}_{\pm 29.3}$ | $\mathbf{44.1}_{\pm 3.0}$ | $\mathbf{241.7}_{\pm 24.8}$ | $\mathbf{18.1}_{\pm 1.3}$ | $\mathbf{105.0}_{\pm 10.2}$ |

(a) TIM classification task

| OOD | STL-10 | | CIFAR-10 | | CIFAR-100 | |
|---|---|---|---|---|---|---|
| | Miss | Correct | Miss | Correct | Miss | Correct |
| $DPN_{rev}$ | $9.0_{\pm 0.6}$ | $9.2_{\pm 1.0}$ | $2.3_{\pm 0.4}$ | $3.2_{\pm 0.5}$ | $2.3_{\pm 0.4}$ | $3.1_{\pm 0.5}$ |
| $DPN^{+}$ | $21.1_{\pm 3.4}$ | $27.1_{\pm 3.9}$ | $17.7_{\pm 3.0}$ | $23.5_{\pm 3.5}$ | $17.3_{\pm 3.0}$ | $23.1_{\pm 3.5}$ |
| $DPN^{-}$ | $\mathbf{729.8}_{\pm 12.4}$ | $\mathbf{1360.4}_{\pm 94.5}$ | $\mathbf{664.9}_{\pm 11.6}$ | $\mathbf{1241.4}_{\pm 79.8}$ | $\mathbf{606.9}_{\pm 11.4}$ | $\mathbf{1134.5}_{\pm 72.1}$ |

| OOD | LSUN | | Places365 | | Textures | |
|---|---|---|---|---|---|---|
| | Miss | Correct | Miss | Correct | Miss | Correct |
| $DPN_{rev}$ | $9.7_{\pm 0.6}$ | $9.9_{\pm 1.0}$ | $8.3_{\pm 0.7}$ | $8.5_{\pm 1.0}$ | $4.4_{\pm 0.4}$ | $4.8_{\pm 0.7}$ |
| $DPN^{+}$ | $21.5_{\pm 3.4}$ | $27.5_{\pm 3.9}$ | $20.6_{\pm 3.3}$ | $26.5_{\pm 3.8}$ | $14.7_{\pm 2.6}$ | $20.0_{\pm 3.0}$ |
| $DPN^{-}$ | $\mathbf{731.6}_{\pm 12.0}$ | $\mathbf{1363.4}_{\pm 91.5}$ | $\mathbf{713.2}_{\pm 11.7}$ | $\mathbf{1330.5}_{\pm 89.2}$ | $\mathbf{465.7}_{\pm 8.0}$ | $\mathbf{873.2}_{\pm 62.7}$ |

Table 11: Results of OOD detection for C10. We report (mean ± standard deviation) of three different models. Description of these OOD datasets are provided in Appendix B.2.1.

| | Methods | AUROC | | | | | AUPR | | | | |
|---|---|---|---|---|---|---|---|---|---|---|---|
| | | Max.P | Ent. | MI | $\alpha_0$ | D-Ent | Max.P | Ent. | MI | $\alpha_0$ | D-Ent |
| Tiny | Baseline | $88.9_{\pm 0.0}$ | $89.5_{\pm 0.0}$ | - | - | - | $85.0_{\pm 0.1}$ | $86.7_{\pm 0.1}$ | - | - | - |
| | MCDP | $88.7_{\pm 0.1}$ | $88.9_{\pm 0.0}$ | $88.1_{\pm 0.1}$ | - | - | $85.2_{\pm 0.1}$ | $85.9_{\pm 0.1}$ | $84.0_{\pm 0.1}$ | - | - |
| | DE | $88.9_{\pm \text{NA}}$ | $89.0_{\pm \text{NA}}$ | $87.8_{\pm \text{NA}}$ | - | - | $85.0_{\pm \text{NA}}$ | $85.5_{\pm \text{NA}}$ | $83.2_{\pm \text{NA}}$ | - | - |
| | EDL | $87.6_{\pm 0.1}$ | $88.4_{\pm 0.2}$ | $89.1_{\pm 0.3}$ | $87.6_{\pm 0.4}$ | $89.9_{\pm 0.2}$ | $84.9_{\pm 0.1}$ | $86.6_{\pm 0.1}$ | $88.9_{\pm 0.3}$ | $87.4_{\pm 0.4}$ | $89.0_{\pm 0.1}$ |
| | OE | $98.2_{\pm 0.1}$ | $98.3_{\pm 0.1}$ | - | - | - | $98.3_{\pm 0.2}$ | $98.3_{\pm 0.2}$ | - | - | - |
| | DPN$_{fwd}$ | $92.8_{\pm 1.0}$ | $93.0_{\pm 1.0}$ | $73.3_{\pm 0.3}$ | $71.3_{\pm 0.4}$ | $93.7_{\pm 1.0}$ | $92.6_{\pm 1.1}$ | $92.9_{\pm 1.1}$ | $59.8_{\pm 0.7}$ | $58.1_{\pm 0.8}$ | $93.4_{\pm 1.3}$ |
| | DPN$_{rev}$ | $97.5_{\pm 0.5}$ | $97.6_{\pm 0.5}$ | $97.8_{\pm 0.4}$ | $97.8_{\pm 0.4}$ | $97.7_{\pm 0.4}$ | $97.5_{\pm 0.5}$ | $97.6_{\pm 0.4}$ | $97.6_{\pm 0.3}$ | $97.6_{\pm 0.3}$ | $97.7_{\pm 0.4}$ |
| | DPN$^+$ | $98.0_{\pm 0.2}$ | $98.0_{\pm 0.2}$ | $98.0_{\pm 0.2}$ | $98.0_{\pm 0.2}$ | $98.0_{\pm 0.2}$ | $98.0_{\pm 0.3}$ | $98.0_{\pm 0.3}$ | $97.9_{\pm 0.3}$ | $97.9_{\pm 0.3}$ | $97.9_{\pm 0.3}$ |
| | DPN$^-$ | $\mathbf{99.0_{\pm 0.1}}$ | $\mathbf{99.0_{\pm 0.1}}$ | $\mathbf{99.0_{\pm 0.1}}$ | $97.7_{\pm 0.1}$ | $6.0_{\pm 0.3}$ | $\mathbf{99.0_{\pm 0.1}}$ | $\mathbf{99.1_{\pm 0.1}}$ | $98.9_{\pm 0.1}$ | $94.9_{\pm 0.1}$ | $32.4_{\pm 0.1}$ |
| STL-10 | Baseline | $75.9_{\pm 0.0}$ | $76.2_{\pm 0.0}$ | - | - | - | $68.5_{\pm 0.1}$ | $69.8_{\pm 0.1}$ | - | - | - |
| | MCDP | $76.2_{\pm 0.0}$ | $76.2_{\pm 0.0}$ | $76.0_{\pm 0.0}$ | - | - | $69.3_{\pm 0.0}$ | $69.7_{\pm 0.0}$ | $69.4_{\pm 0.1}$ | - | - |
| | DE | $76.0_{\pm \text{NA}}$ | $76.0_{\pm \text{NA}}$ | $75.6_{\pm \text{NA}}$ | - | - | $68.5_{\pm \text{NA}}$ | $68.9_{\pm \text{NA}}$ | $67.6_{\pm \text{NA}}$ | - | - |
| | EDL | $72.4_{\pm 0.1}$ | $72.7_{\pm 0.1}$ | $71.7_{\pm 0.1}$ | $70.6_{\pm 0.1}$ | $73.3_{\pm 0.1}$ | $66.8_{\pm 0.0}$ | $68.0_{\pm 0.0}$ | $68.2_{\pm 0.2}$ | $66.7_{\pm 0.3}$ | $69.4_{\pm 0.0}$ |
| | OE | $81.4_{\pm 1.2}$ | $81.5_{\pm 1.2}$ | - | - | - | $80.8_{\pm 1.1}$ | $80.8_{\pm 1.0}$ | - | - | - |
| | DPN$_{fwd}$ | $71.5_{\pm 1.3}$ | $71.6_{\pm 1.3}$ | $65.7_{\pm 0.2}$ | $64.9_{\pm 0.2}$ | $72.0_{\pm 1.5}$ | $68.2_{\pm 2.1}$ | $68.6_{\pm 2.1}$ | $53.3_{\pm 0.6}$ | $52.5_{\pm 0.7}$ | $68.6_{\pm 2.7}$ |
| | DPN$_{rev}$ | $81.6_{\pm 1.7}$ | $81.7_{\pm 1.7}$ | $82.2_{\pm 1.7}$ | $82.2_{\pm 1.6}$ | $81.9_{\pm 1.7}$ | $81.9_{\pm 1.7}$ | $82.0_{\pm 1.7}$ | $82.5_{\pm 1.7}$ | $82.5_{\pm 1.6}$ | $82.2_{\pm 1.7}$ |
| | DPN$^+$ | $81.6_{\pm 1.4}$ | $81.7_{\pm 1.3}$ | $81.8_{\pm 1.2}$ | $81.8_{\pm 1.2}$ | $81.8_{\pm 1.2}$ | $80.9_{\pm 1.2}$ | $81.0_{\pm 1.3}$ | $81.0_{\pm 1.2}$ | $81.0_{\pm 1.2}$ | $81.0_{\pm 1.2}$ |
| | DPN$^-$ | $84.7_{\pm 0.4}$ | $84.8_{\pm 0.5}$ | $\mathbf{85.3_{\pm 0.5}}$ | $84.9_{\pm 0.5}$ | $34.6_{\pm 0.4}$ | $84.7_{\pm 0.6}$ | $84.9_{\pm 0.6}$ | $\mathbf{85.2_{\pm 0.6}}$ | $82.0_{\pm 0.6}$ | $42.4_{\pm 0.2}$ |
| LSUN | Baseline | $90.3_{\pm 0.0}$ | $91.0_{\pm 0.0}$ | - | - | - | $86.6_{\pm 0.1}$ | $88.5_{\pm 0.1}$ | - | - | - |
| | MCDP | $90.6_{\pm 0.0}$ | $90.8_{\pm 0.0}$ | $90.2_{\pm 0.0}$ | - | - | $87.5_{\pm 0.1}$ | $88.2_{\pm 0.0}$ | $86.6_{\pm 0.1}$ | - | - |
| | DE | $90.3_{\pm \text{NA}}$ | $90.4_{\pm \text{NA}}$ | $89.7_{\pm \text{NA}}$ | - | - | $86.5_{\pm \text{NA}}$ | $87.1_{\pm \text{NA}}$ | $85.6_{\pm \text{NA}}$ | - | - |
| | EDL | $90.3_{\pm 0.0}$ | $91.2_{\pm 0.0}$ | $93.5_{\pm 0.0}$ | $92.6_{\pm 0.0}$ | $93.0_{\pm 0.0}$ | $87.8_{\pm 0.0}$ | $89.7_{\pm 0.0}$ | $93.4_{\pm 0.0}$ | $92.3_{\pm 0.0}$ | $92.4_{\pm 0.0}$ |
| | OE | $98.4_{\pm 0.3}$ | $98.4_{\pm 0.3}$ | - | - | - | $98.2_{\pm 0.4}$ | $98.2_{\pm 0.4}$ | - | - | - |
| | DPN$_{fwd}$ | $93.5_{\pm 0.7}$ | $93.7_{\pm 0.8}$ | $72.6_{\pm 0.1}$ | $70.6_{\pm 0.2}$ | $94.9_{\pm 0.8}$ | $93.2_{\pm 0.9}$ | $93.5_{\pm 0.9}$ | $58.6_{\pm 0.3}$ | $57.0_{\pm 0.3}$ | $94.4_{\pm 1.0}$ |
| | DPN$_{rev}$ | $98.5_{\pm 0.4}$ | $98.6_{\pm 0.3}$ | $98.7_{\pm 0.3}$ | $98.7_{\pm 0.3}$ | $98.7_{\pm 0.3}$ | $98.3_{\pm 0.4}$ | $98.4_{\pm 0.3}$ | $98.5_{\pm 0.3}$ | $98.5_{\pm 0.3}$ | $98.5_{\pm 0.3}$ |
| | DPN$^+$ | $98.2_{\pm 0.3}$ | $98.3_{\pm 0.4}$ | $98.3_{\pm 0.4}$ | $98.3_{\pm 0.4}$ | $98.3_{\pm 0.4}$ | $98.0_{\pm 0.5}$ | $98.1_{\pm 0.5}$ | $98.0_{\pm 0.4}$ | $98.0_{\pm 0.4}$ | $98.0_{\pm 0.4}$ |
| | DPN$^-$ | $99.2_{\pm 0.1}$ | $99.2_{\pm 0.1}$ | $\mathbf{99.3_{\pm 0.0}}$ | $98.1_{\pm 0.1}$ | $5.0_{\pm 0.2}$ | $99.1_{\pm 0.1}$ | $\mathbf{99.2_{\pm 0.1}}$ | $99.1_{\pm 0.1}$ | $95.9_{\pm 0.0}$ | $32.1_{\pm 0.2}$ |
| Places365 | Baseline | $89.4_{\pm 0.0}$ | $90.0_{\pm 0.0}$ | - | - | - | $95.5_{\pm 0.0}$ | $96.1_{\pm 0.0}$ | - | - | - |
| | MCDP | $89.5_{\pm 0.0}$ | $89.7_{\pm 0.0}$ | $89.0_{\pm 0.0}$ | - | - | $95.7_{\pm 0.0}$ | $96.0_{\pm 0.0}$ | $95.3_{\pm 0.0}$ | - | - |
| | DE | $89.4_{\pm \text{NA}}$ | $89.5_{\pm \text{NA}}$ | $88.7_{\pm \text{NA}}$ | - | - | $95.5_{\pm \text{NA}}$ | $95.7_{\pm \text{NA}}$ | $95.2_{\pm \text{NA}}$ | - | - |
| | EDL | $88.5_{\pm 0.0}$ | $89.4_{\pm 0.0}$ | $91.3_{\pm 0.0}$ | $90.3_{\pm 0.1}$ | $91.2_{\pm 0.0}$ | $95.6_{\pm 0.0}$ | $96.2_{\pm 0.0}$ | $97.3_{\pm 0.0}$ | $96.8_{\pm 0.0}$ | $97.1_{\pm 0.0}$ |
| | OE | $98.1_{\pm 0.1}$ | $98.2_{\pm 0.2}$ | - | - | - | $99.4_{\pm 0.1}$ | $99.4_{\pm 0.1}$ | - | - | - |
| | DPN$_{fwd}$ | $93.3_{\pm 0.9}$ | $93.5_{\pm 0.8}$ | $72.1_{\pm 0.2}$ | $70.1_{\pm 0.3}$ | $94.4_{\pm 0.9}$ | $97.9_{\pm 0.3}$ | $98.0_{\pm 0.3}$ | $82.4_{\pm 0.6}$ | $81.4_{\pm 0.6}$ | $98.2_{\pm 0.3}$ |
| | DPN$_{rev}$ | $97.8_{\pm 0.4}$ | $97.9_{\pm 0.4}$ | $98.0_{\pm 0.4}$ | $98.0_{\pm 0.4}$ | $98.0_{\pm 0.4}$ | $99.3_{\pm 0.1}$ | $99.3_{\pm 0.1}$ | $99.4_{\pm 0.1}$ | $99.4_{\pm 0.1}$ | $99.4_{\pm 0.1}$ |
| | DPN$^+$ | $98.0_{\pm 0.3}$ | $98.0_{\pm 0.3}$ | $98.0_{\pm 0.3}$ | $98.0_{\pm 0.3}$ | $98.0_{\pm 0.3}$ | $99.4_{\pm 0.1}$ | $99.4_{\pm 0.1}$ | $99.4_{\pm 0.1}$ | $99.4_{\pm 0.1}$ | $99.4_{\pm 0.1}$ |
| | DPN$^-$ | $\mathbf{98.9_{\pm 0.1}}$ | $\mathbf{98.9_{\pm 0.1}}$ | $98.9_{\pm 0.1}$ | $97.7_{\pm 0.1}$ | $6.2_{\pm 0.4}$ | $\mathbf{99.7_{\pm 0.0}}$ | $\mathbf{99.7_{\pm 0.0}}$ | $\mathbf{99.7_{\pm 0.0}}$ | $98.6_{\pm 0.0}$ | $61.0_{\pm 0.2}$ |
| Textures | Baseline | $88.8_{\pm 0.0}$ | $89.2_{\pm 0.0}$ | - | - | - | $74.9_{\pm 0.1}$ | $76.9_{\pm 0.2}$ | - | - | - |
| | MCDP | $87.4_{\pm 0.1}$ | $87.5_{\pm 0.1}$ | $85.7_{\pm 0.2}$ | - | - | $73.3_{\pm 0.2}$ | $74.2_{\pm 0.1}$ | $66.9_{\pm 0.3}$ | - | - |
| | DE | $88.7_{\pm \text{NA}}$ | $88.8_{\pm \text{NA}}$ | $87.2_{\pm \text{NA}}$ | - | - | $74.8_{\pm \text{NA}}$ | $75.4_{\pm \text{NA}}$ | $72.1_{\pm \text{NA}}$ | - | - |
| | EDL | $84.3_{\pm 0.4}$ | $84.9_{\pm 0.5}$ | $84.9_{\pm 0.9}$ | $83.1_{\pm 1.0}$ | $86.3_{\pm 0.6}$ | $71.5_{\pm 0.4}$ | $73.6_{\pm 0.5}$ | $76.4_{\pm 1.1}$ | $73.9_{\pm 1.3}$ | $77.0_{\pm 0.6}$ |
| | OE | $99.4_{\pm 0.0}$ | $99.4_{\pm 0.1}$ | - | - | - | $98.9_{\pm 0.2}$ | $98.9_{\pm 0.3}$ | - | - | - |
| | DPN$_{fwd}$ | $98.3_{\pm 0.4}$ | $98.4_{\pm 0.4}$ | $68.2_{\pm 0.2}$ | $65.8_{\pm 0.1}$ | $98.6_{\pm 0.2}$ | $97.1_{\pm 0.6}$ | $97.4_{\pm 0.6}$ | $42.0_{\pm 0.2}$ | $40.4_{\pm 0.2}$ | $97.7_{\pm 0.6}$ |
| | DPN$_{rev}$ | $99.4_{\pm 0.0}$ | $99.3_{\pm 0.0}$ | $99.4_{\pm 0.0}$ | $99.4_{\pm 0.0}$ | $99.4_{\pm 0.0}$ | $98.5_{\pm 0.0}$ | $98.3_{\pm 0.0}$ | $98.5_{\pm 0.1}$ | $98.4_{\pm 0.1}$ | $98.4_{\pm 0.0}$ |
| | DPN$^+$ | $99.5_{\pm 0.0}$ | $99.6_{\pm 0.0}$ | $99.5_{\pm 0.0}$ | $99.5_{\pm 0.0}$ | $99.5_{\pm 0.0}$ | $99.1_{\pm 0.2}$ | $99.1_{\pm 0.1}$ | $99.1_{\pm 0.1}$ | $99.1_{\pm 0.1}$ | $99.1_{\pm 0.1}$ |
| | DPN$^-$ | $\mathbf{99.7_{\pm 0.0}}$ | $\mathbf{99.7_{\pm 0.0}}$ | $99.5_{\pm 0.0}$ | $97.8_{\pm 0.1}$ | $3.6_{\pm 0.1}$ | $\mathbf{99.4_{\pm 0.0}}$ | $\mathbf{99.4_{\pm 0.1}}$ | $98.8_{\pm 0.0}$ | $90.1_{\pm 0.3}$ | $21.3_{\pm 0.0}$ |

Table 12: Results of OOD image detection for C100. We report (mean ± standard deviation) of three different models. Description of these OOD datasets are provided in Appendix B.2.1.

| | Methods | AUROC | | | | | AUPR | | | | |
|---|---|---|---|---|---|---|---|---|---|---|---|
| | | Max.P | Ent. | MI | $\alpha_0$ | D-Ent | Max.P | Ent. | MI | $\alpha_0$ | D-Ent |
| Tiny | Baseline | $68.8_{\pm0.2}$ | $71.4_{\pm0.2}$ | - | - | - | $66.6_{\pm0.2}$ | $70.2_{\pm0.2}$ | - | - | - |
| | MCDP | $69.7_{\pm0.3}$ | $70.2_{\pm0.3}$ | $70.6_{\pm0.3}$ | - | - | $67.4_{\pm0.3}$ | $68.5_{\pm0.2}$ | $66.0_{\pm0.2}$ | - | - |
| | DE | $68.9_{\pm\text{NA}}$ | $69.3_{\pm\text{NA}}$ | $69.6_{\pm\text{NA}}$ | - | - | $66.7_{\pm\text{NA}}$ | $67.7_{\pm\text{NA}}$ | $66.3_{\pm\text{NA}}$ | - | - |
| | EDL | $66.9_{\pm0.2}$ | $71.5_{\pm0.2}$ | $72.8_{\pm0.5}$ | $72.2_{\pm0.6}$ | $77.4_{\pm0.1}$ | $62.8_{\pm0.4}$ | $68.9_{\pm0.6}$ | $71.4_{\pm0.7}$ | $71.0_{\pm0.7}$ | $74.8_{\pm0.4}$ |
| | OE | $89.5_{\pm1.0}$ | $91.2_{\pm0.9}$ | - | - | - | $91.1_{\pm0.9}$ | $92.6_{\pm0.8}$ | - | - | - |
| | DPN$_{rev}$ | $81.2_{\pm0.2}$ | $82.4_{\pm0.1}$ | $83.8_{\pm0.1}$ | $83.8_{\pm0.1}$ | $83.5_{\pm0.1}$ | $84.7_{\pm0.0}$ | $86.1_{\pm0.0}$ | $87.6_{\pm0.0}$ | $87.6_{\pm0.0}$ | $87.1_{\pm0.0}$ |
| | DPN$^+$ | $85.9_{\pm0.3}$ | $88.1_{\pm0.2}$ | $92.2_{\pm0.1}$ | $92.2_{\pm0.1}$ | $92.3_{\pm0.1}$ | $88.0_{\pm0.2}$ | $90.0_{\pm0.2}$ | $92.7_{\pm0.1}$ | $92.7_{\pm0.1}$ | $92.8_{\pm0.1}$ |
| | DPN$^-$ | $89.2_{\pm0.1}$ | $90.7_{\pm0.1}$ | $\mathbf{94.5_{\pm0.1}}$ | $\mathbf{94.5_{\pm0.1}}$ | $38.1_{\pm0.5}$ | $91.4_{\pm0.1}$ | $92.7_{\pm0.1}$ | $\mathbf{95.1_{\pm0.1}}$ | $\mathbf{95.1_{\pm0.1}}$ | $55.7_{\pm0.4}$ |
| STL-10 | Baseline | $69.6_{\pm0.0}$ | $71.9_{\pm0.0}$ | - | - | - | $61.9_{\pm0.1}$ | $65.4_{\pm0.1}$ | - | - | - |
| | MCDP | $70.7_{\pm0.1}$ | $71.2_{\pm0.1}$ | $71.6_{\pm0.2}$ | - | - | $62.8_{\pm0.1}$ | $63.9_{\pm0.2}$ | $61.4_{\pm0.1}$ | - | - |
| | DE | $69.6_{\pm\text{NA}}$ | $70.1_{\pm\text{NA}}$ | $70.2_{\pm\text{NA}}$ | - | - | $62.0_{\pm\text{NA}}$ | $63.0_{\pm\text{NA}}$ | $60.9_{\pm\text{NA}}$ | - | - |
| | EDL | $68.1_{\pm0.2}$ | $72.0_{\pm0.2}$ | $68.0_{\pm0.6}$ | $67.3_{\pm0.7}$ | $73.8_{\pm0.4}$ | $58.5_{\pm0.6}$ | $64.1_{\pm0.4}$ | $61.6_{\pm1.0}$ | $61.1_{\pm1.1}$ | $66.1_{\pm0.7}$ |
| | OE | $91.2_{\pm0.7}$ | $92.7_{\pm0.6}$ | - | - | - | $92.1_{\pm0.6}$ | $93.4_{\pm0.5}$ | - | - | - |
| | DPN$_{rev}$ | $87.2_{\pm0.1}$ | $88.1_{\pm0.1}$ | $89.3_{\pm0.1}$ | $89.3_{\pm0.1}$ | $89.0_{\pm0.1}$ | $88.5_{\pm0.0}$ | $89.6_{\pm0.1}$ | $91.0_{\pm0.1}$ | $91.1_{\pm0.1}$ | $90.5_{\pm0.1}$ |
| | DPN$^+$ | $89.1_{\pm0.2}$ | $90.8_{\pm0.2}$ | $95.0_{\pm0.0}$ | $95.0_{\pm0.0}$ | $94.8_{\pm0.0}$ | $90.0_{\pm0.2}$ | $91.7_{\pm0.2}$ | $94.7_{\pm0.0}$ | $94.7_{\pm0.0}$ | $94.6_{\pm0.1}$ |
| | DPN$^-$ | $92.8_{\pm0.1}$ | $93.9_{\pm0.1}$ | $\mathbf{96.8_{\pm0.1}}$ | $\mathbf{96.8_{\pm0.1}}$ | $25.4_{\pm0.4}$ | $93.7_{\pm0.1}$ | $94.7_{\pm0.1}$ | $\mathbf{96.7_{\pm0.1}}$ | $\mathbf{96.7_{\pm0.1}}$ | $42.8_{\pm0.3}$ |
| LSUN | Baseline | $72.5_{\pm0.0}$ | $75.0_{\pm0.0}$ | - | - | - | $69.0_{\pm0.1}$ | $72.7_{\pm0.1}$ | - | - | - |
| | MCDP | $74.5_{\pm0.1}$ | $75.1_{\pm0.1}$ | $75.9_{\pm0.2}$ | - | - | $70.8_{\pm0.3}$ | $71.9_{\pm0.2}$ | $70.4_{\pm0.2}$ | - | - |
| | DE | $72.6_{\pm\text{NA}}$ | $73.0_{\pm\text{NA}}$ | $73.4_{\pm\text{NA}}$ | - | - | $69.1_{\pm\text{NA}}$ | $70.0_{\pm\text{NA}}$ | $68.6_{\pm\text{NA}}$ | - | - |
| | EDL | $67.6_{\pm0.6}$ | $72.3_{\pm0.6}$ | $72.8_{\pm0.6}$ | $72.3_{\pm0.6}$ | $76.7_{\pm0.5}$ | $62.3_{\pm1.0}$ | $69.3_{\pm1.1}$ | $72.9_{\pm0.8}$ | $72.5_{\pm0.9}$ | $76.0_{\pm0.5}$ |
| | OE | $92.2_{\pm0.9}$ | $93.7_{\pm0.7}$ | - | - | - | $93.7_{\pm0.7}$ | $94.9_{\pm0.7}$ | - | - | - |
| | DPN$_{rev}$ | $86.7_{\pm0.0}$ | $87.9_{\pm0.0}$ | $89.3_{\pm0.1}$ | $89.3_{\pm0.1}$ | $88.9_{\pm0.1}$ | $89.2_{\pm0.0}$ | $90.5_{\pm0.0}$ | $92.0_{\pm0.0}$ | $92.0_{\pm0.0}$ | $91.5_{\pm0.0}$ |
| | DPN$^+$ | $90.3_{\pm0.3}$ | $92.1_{\pm0.3}$ | $95.0_{\pm0.1}$ | $95.0_{\pm0.1}$ | $95.0_{\pm0.1}$ | $92.0_{\pm0.2}$ | $93.6_{\pm0.2}$ | $95.5_{\pm0.1}$ | $95.5_{\pm0.1}$ | $95.6_{\pm0.1}$ |
| | DPN$^-$ | $92.8_{\pm0.1}$ | $94.0_{\pm0.1}$ | $\mathbf{96.5_{\pm0.1}}$ | $\mathbf{96.5_{\pm0.1}}$ | $31.5_{\pm0.4}$ | $94.3_{\pm0.1}$ | $95.3_{\pm0.1}$ | $\mathbf{97.0_{\pm0.1}}$ | $96.9_{\pm0.1}$ | $52.6_{\pm0.3}$ |
| Places365 | Baseline | $70.7_{\pm0.0}$ | $73.2_{\pm0.0}$ | - | - | - | $88.0_{\pm0.0}$ | $89.6_{\pm0.0}$ | - | - | - |
| | MCDP | $72.1_{\pm0.1}$ | $72.6_{\pm0.1}$ | $73.4_{\pm0.2}$ | - | - | $88.6_{\pm0.0}$ | $89.0_{\pm0.0}$ | $88.4_{\pm0.1}$ | - | - |
| | DE | $70.8_{\pm\text{NA}}$ | $71.2_{\pm\text{NA}}$ | $72.3_{\pm\text{NA}}$ | - | - | $88.1_{\pm\text{NA}}$ | $88.4_{\pm\text{NA}}$ | $88.6_{\pm\text{NA}}$ | - | - |
| | EDL | $66.8_{\pm0.5}$ | $71.0_{\pm0.6}$ | $70.2_{\pm0.3}$ | $69.7_{\pm0.3}$ | $74.3_{\pm0.4}$ | $85.5_{\pm0.5}$ | $88.4_{\pm0.4}$ | $89.2_{\pm0.2}$ | $89.0_{\pm0.3}$ | $90.6_{\pm0.1}$ |
| | OE | $89.3_{\pm1.0}$ | $90.9_{\pm0.9}$ | - | - | - | $97.0_{\pm0.3}$ | $97.5_{\pm0.3}$ | - | - | - |
| | DPN$_{rev}$ | $83.0_{\pm0.1}$ | $84.2_{\pm0.1}$ | $85.8_{\pm0.1}$ | $85.8_{\pm0.1}$ | $85.4_{\pm0.1}$ | $95.1_{\pm0.0}$ | $95.6_{\pm0.0}$ | $96.1_{\pm0.0}$ | $96.1_{\pm0.0}$ | $95.9_{\pm0.0}$ |
| | DPN$^+$ | $87.1_{\pm0.2}$ | $89.3_{\pm0.2}$ | $92.9_{\pm0.1}$ | $92.9_{\pm0.1}$ | $93.0_{\pm0.1}$ | $96.3_{\pm0.0}$ | $96.9_{\pm0.1}$ | $97.9_{\pm0.0}$ | $97.9_{\pm0.0}$ | $97.9_{\pm0.0}$ |
| | DPN$^-$ | $89.9_{\pm0.2}$ | $91.3_{\pm0.1}$ | $\mathbf{94.5_{\pm0.1}}$ | $\mathbf{94.5_{\pm0.1}}$ | $37.9_{\pm0.5}$ | $97.2_{\pm0.0}$ | $97.6_{\pm0.0}$ | $\mathbf{98.4_{\pm0.0}}$ | $\mathbf{98.4_{\pm0.0}}$ | $79.8_{\pm0.2}$ |
| Textures | Baseline | $62.8_{\pm0.2}$ | $64.7_{\pm0.2}$ | - | - | - | $43.8_{\pm0.2}$ | $45.9_{\pm0.2}$ | - | - | - |
| | MCDP | $64.3_{\pm0.3}$ | $64.6_{\pm0.2}$ | $67.2_{\pm0.2}$ | - | - | $44.9_{\pm0.3}$ | $45.3_{\pm0.2}$ | $49.6_{\pm0.0}$ | - | - |
| | DE | $62.9_{\pm\text{NA}}$ | $63.0_{\pm\text{NA}}$ | $69.0_{\pm\text{NA}}$ | - | - | $43.9_{\pm\text{NA}}$ | $44.1_{\pm\text{NA}}$ | $56.1_{\pm\text{NA}}$ | - | - |
| | EDL | $66.9_{\pm0.7}$ | $72.1_{\pm0.8}$ | $73.7_{\pm0.5}$ | $73.1_{\pm0.4}$ | $78.1_{\pm1.4}$ | $47.5_{\pm2.2}$ | $55.5_{\pm2.4}$ | $60.1_{\pm1.0}$ | $59.5_{\pm0.9}$ | $64.8_{\pm1.4}$ |
| | OE | $79.7_{\pm1.0}$ | $81.2_{\pm1.0}$ | - | - | - | $71.7_{\pm1.5}$ | $73.2_{\pm1.6}$ | - | - | - |
| | DPN$_{rev}$ | $73.7_{\pm0.5}$ | $75.3_{\pm0.4}$ | $76.9_{\pm0.4}$ | $76.9_{\pm0.4}$ | $76.8_{\pm0.4}$ | $67.3_{\pm0.4}$ | $70.5_{\pm0.3}$ | $73.2_{\pm0.3}$ | $73.1_{\pm0.3}$ | $72.9_{\pm0.3}$ |
| | DPN$^+$ | $78.8_{\pm0.1}$ | $81.3_{\pm0.1}$ | $83.6_{\pm0.1}$ | $83.6_{\pm0.1}$ | $84.8_{\pm0.1}$ | $68.5_{\pm0.2}$ | $71.4_{\pm0.1}$ | $71.8_{\pm0.1}$ | $71.8_{\pm0.1}$ | $73.3_{\pm0.1}$ |
| | DPN$^-$ | $77.5_{\pm0.0}$ | $79.5_{\pm0.0}$ | $\mathbf{85.2_{\pm0.1}}$ | $\mathbf{85.2_{\pm0.1}}$ | $58.4_{\pm0.2}$ | $71.9_{\pm0.1}$ | $74.4_{\pm0.1}$ | $\mathbf{78.9_{\pm0.2}}$ | $\mathbf{78.9_{\pm0.2}}$ | $52.1_{\pm0.2}$ |

Table 13: Results of OOD image detection for TIM. We report (mean ± standard deviation) of three different models. Description of these OOD datasets are provided in Appendix B.2.1.

| Dataset | Methods | AUROC | | | | | AUPR | | | | |
|---|---|---|---|---|---|---|---|---|---|---|---|
| | | Max.P | Ent. | MI | $\alpha_0$ | D-Ent | Max.P | Ent. | MI | $\alpha_0$ | D-Ent |
| CIFAR-10 | Baseline | $76.9_{\pm0.2}$ | $79.2_{\pm0.2}$ | - | - | - | $72.6_{\pm0.2}$ | $75.1_{\pm0.1}$ | - | - | - |
| | MCDP | $77.4_{\pm0.1}$ | $79.8_{\pm0.1}$ | $77.5_{\pm0.2}$ | - | - | $73.0_{\pm0.2}$ | $75.4_{\pm0.1}$ | $71.4_{\pm0.4}$ | - | - |
| | DE | $76.9_{\pm\text{NA}}$ | $79.3_{\pm\text{NA}}$ | $77.7_{\pm\text{NA}}$ | - | - | $72.6_{\pm\text{NA}}$ | $75.2_{\pm\text{NA}}$ | $72.4_{\pm\text{NA}}$ | - | - |
| | EDL | $74.4_{\pm0.7}$ | $76.1_{\pm0.7}$ | $72.0_{\pm1.3}$ | $71.8_{\pm1.3}$ | $73.3_{\pm1.1}$ | $69.4_{\pm0.3}$ | $70.9_{\pm0.8}$ | $67.8_{\pm1.3}$ | $67.7_{\pm1.4}$ | $68.7_{\pm1.3}$ |
| | OE | $91.3_{\pm0.4}$ | $92.6_{\pm0.3}$ | - | - | - | $92.9_{\pm0.4}$ | $94.0_{\pm0.3}$ | - | - | - |
| | $DPN_{rev}$ | $85.4_{\pm0.7}$ | $86.0_{\pm0.8}$ | $82.8_{\pm1.4}$ | $81.9_{\pm1.6}$ | $85.6_{\pm0.9}$ | $88.1_{\pm0.7}$ | $88.7_{\pm0.7}$ | $86.5_{\pm1.3}$ | $86.0_{\pm1.4}$ | $88.7_{\pm0.8}$ |
| | $DPN^+$ | $99.2_{\pm0.0}$ | $99.4_{\pm0.0}$ | $99.7_{\pm0.0}$ | $99.7_{\pm0.0}$ | $99.6_{\pm0.0}$ | $99.4_{\pm0.0}$ | $99.5_{\pm0.0}$ | $99.7_{\pm0.0}$ | $99.7_{\pm0.0}$ | $99.6_{\pm0.0}$ |
| | $DPN^-$ | $99.7_{\pm0.0}$ | $99.8_{\pm0.0}$ | $\mathbf{99.9_{\pm0.0}}$ | $\mathbf{99.9_{\pm0.0}}$ | $3.5_{\pm0.1}$ | $99.8_{\pm0.0}$ | $99.8_{\pm0.0}$ | $\mathbf{99.9_{\pm0.0}}$ | $\mathbf{99.9_{\pm0.0}}$ | $33.6_{\pm0.1}$ |
| CIFAR-100 | Baseline | $73.6_{\pm0.2}$ | $75.7_{\pm0.2}$ | - | - | - | $69.7_{\pm0.2}$ | $72.2_{\pm0.2}$ | - | - | - |
| | MCDP | $74.0_{\pm0.2}$ | $76.1_{\pm0.2}$ | $73.6_{\pm0.2}$ | - | - | $70.0_{\pm0.1}$ | $72.3_{\pm0.1}$ | $67.6_{\pm0.3}$ | - | - |
| | DE | $73.7_{\pm\text{NA}}$ | $75.8_{\pm\text{NA}}$ | $75.3_{\pm\text{NA}}$ | - | - | $69.7_{\pm\text{NA}}$ | $72.3_{\pm\text{NA}}$ | $70.6_{\pm\text{NA}}$ | - | - |
| | EDL | $71.8_{\pm0.6}$ | $73.7_{\pm0.7}$ | $70.1_{\pm0.9}$ | $69.9_{\pm0.9}$ | $71.4_{\pm0.8}$ | $67.2_{\pm0.4}$ | $68.6_{\pm0.9}$ | $65.9_{\pm1.0}$ | $65.8_{\pm1.0}$ | $66.7_{\pm1.0}$ |
| | OE | $89.5_{\pm0.5}$ | $90.9_{\pm0.4}$ | - | - | - | $91.5_{\pm0.5}$ | $92.7_{\pm0.4}$ | - | - | - |
| | $DPN_{rev}$ | $84.2_{\pm0.8}$ | $85.0_{\pm0.8}$ | $82.5_{\pm1.4}$ | $81.7_{\pm1.6}$ | $85.0_{\pm0.9}$ | $87.0_{\pm0.7}$ | $87.9_{\pm0.7}$ | $86.3_{\pm1.3}$ | $85.7_{\pm1.4}$ | $88.2_{\pm0.8}$ |
| | $DPN^+$ | $98.8_{\pm0.0}$ | $99.0_{\pm0.0}$ | $99.5_{\pm0.0}$ | $99.5_{\pm0.0}$ | $99.4_{\pm0.0}$ | $99.1_{\pm0.0}$ | $99.3_{\pm0.0}$ | $99.5_{\pm0.0}$ | $99.5_{\pm0.0}$ | $99.5_{\pm0.0}$ |
| | $DPN^-$ | $98.7_{\pm0.1}$ | $99.0_{\pm0.0}$ | $\mathbf{99.6_{\pm0.0}}$ | $\mathbf{99.6_{\pm0.0}}$ | $7.5_{\pm0.2}$ | $99.1_{\pm0.0}$ | $99.3_{\pm0.0}$ | $\mathbf{99.7_{\pm0.0}}$ | $\mathbf{99.7_{\pm0.0}}$ | $36.7_{\pm0.2}$ |
| STL-10 | Baseline | $67.7_{\pm0.1}$ | $67.7_{\pm0.1}$ | - | - | - | $56.5_{\pm0.1}$ | $56.8_{\pm0.2}$ | - | - | - |
| | MCDP | $68.1_{\pm0.1}$ | $68.1_{\pm0.1}$ | $69.8_{\pm0.1}$ | - | - | $56.8_{\pm0.2}$ | $57.3_{\pm0.2}$ | $60.1_{\pm0.4}$ | - | - |
| | DE | $67.7_{\pm\text{NA}}$ | $67.8_{\pm\text{NA}}$ | $68.1_{\pm\text{NA}}$ | - | - | $56.5_{\pm\text{NA}}$ | $56.8_{\pm\text{NA}}$ | $58.1_{\pm\text{NA}}$ | - | - |
| | EDL | $66.5_{\pm0.3}$ | $65.0_{\pm0.6}$ | $60.4_{\pm1.0}$ | $60.3_{\pm1.0}$ | $61.3_{\pm0.9}$ | $55.7_{\pm0.1}$ | $55.6_{\pm0.4}$ | $53.7_{\pm0.7}$ | $53.6_{\pm0.8}$ | $54.1_{\pm0.7}$ |
| | OE | $\mathbf{100.0_{\pm0.0}}$ | $\mathbf{100.0_{\pm0.0}}$ | - | - | - | $\mathbf{100.0_{\pm0.0}}$ | $\mathbf{100.0_{\pm0.0}}$ | - | - | - |
| | $DPN_{rev}$ | $99.4_{\pm0.0}$ | $99.5_{\pm0.0}$ | $99.6_{\pm0.0}$ | $99.6_{\pm0.0}$ | $99.5_{\pm0.0}$ | $99.5_{\pm0.0}$ | $99.6_{\pm0.0}$ | $99.7_{\pm0.0}$ | $99.7_{\pm0.0}$ | $99.6_{\pm0.0}$ |
| | $DPN^+$ | $99.9_{\pm0.0}$ | $\mathbf{100.0_{\pm0.0}}$ | $\mathbf{100.0_{\pm0.0}}$ | $\mathbf{100.0_{\pm0.0}}$ | $\mathbf{100.0_{\pm0.0}}$ | $99.9_{\pm0.0}$ | $\mathbf{100.0_{\pm0.0}}$ | $99.9_{\pm0.1}$ | $99.9_{\pm0.1}$ | $99.9_{\pm0.0}$ |
| | $DPN^-$ | $\mathbf{100.0_{\pm0.0}}$ | $\mathbf{100.0_{\pm0.0}}$ | $\mathbf{100.0_{\pm0.0}}$ | $\mathbf{100.0_{\pm0.0}}$ | $0.2_{\pm0.0}$ | $\mathbf{100.0_{\pm0.0}}$ | $\mathbf{100.0_{\pm0.0}}$ | $\mathbf{100.0_{\pm0.0}}$ | $\mathbf{100.0_{\pm0.0}}$ | $26.6_{\pm0.0}$ |
| LSUN | Baseline | $69.0_{\pm0.0}$ | $70.2_{\pm0.1}$ | - | - | - | $63.5_{\pm0.2}$ | $64.0_{\pm0.3}$ | - | - | - |
| | MCDP | $69.3_{\pm0.1}$ | $70.4_{\pm0.1}$ | $69.5_{\pm0.2}$ | - | - | $63.6_{\pm0.2}$ | $64.0_{\pm0.3}$ | $63.2_{\pm0.4}$ | - | - |
| | DE | $69.1_{\pm\text{NA}}$ | $70.2_{\pm\text{NA}}$ | $70.3_{\pm\text{NA}}$ | - | - | $63.5_{\pm\text{NA}}$ | $64.0_{\pm\text{NA}}$ | $65.2_{\pm\text{NA}}$ | - | - |
| | EDL | $65.5_{\pm0.2}$ | $67.1_{\pm0.6}$ | $64.7_{\pm1.3}$ | $64.7_{\pm1.3}$ | $65.6_{\pm1.2}$ | $60.4_{\pm0.3}$ | $61.8_{\pm0.8}$ | $60.6_{\pm1.5}$ | $60.5_{\pm1.6}$ | $61.0_{\pm1.4}$ |
| | OE | $\mathbf{100.0_{\pm0.0}}$ | $\mathbf{100.0_{\pm0.0}}$ | - | - | - | $\mathbf{100.0_{\pm0.0}}$ | $\mathbf{100.0_{\pm0.0}}$ | - | - | - |
| | $DPN_{rev}$ | $99.6_{\pm0.0}$ | $99.7_{\pm0.0}$ | $99.8_{\pm0.0}$ | $99.8_{\pm0.0}$ | $99.8_{\pm0.0}$ | $99.7_{\pm0.0}$ | $99.8_{\pm0.0}$ | $99.8_{\pm0.0}$ | $99.8_{\pm0.0}$ | $99.8_{\pm0.0}$ |
| | $DPN^+$ | $\mathbf{100.0_{\pm0.0}}$ | $\mathbf{100.0_{\pm0.0}}$ | $\mathbf{100.0_{\pm0.0}}$ | $\mathbf{100.0_{\pm0.0}}$ | $\mathbf{100.0_{\pm0.0}}$ | $\mathbf{100.0_{\pm0.0}}$ | $\mathbf{100.0_{\pm0.0}}$ | $99.9_{\pm0.0}$ | $99.9_{\pm0.0}$ | $99.9_{\pm0.0}$ |
| | $DPN^-$ | $\mathbf{100.0_{\pm0.0}}$ | $\mathbf{100.0_{\pm0.0}}$ | $\mathbf{100.0_{\pm0.0}}$ | $\mathbf{100.0_{\pm0.0}}$ | $0.2_{\pm0.0}$ | $\mathbf{100.0_{\pm0.0}}$ | $\mathbf{100.0_{\pm0.0}}$ | $\mathbf{100.0_{\pm0.0}}$ | $\mathbf{100.0_{\pm0.0}}$ | $30.8_{\pm0.0}$ |
| Places365 | Baseline | $71.1_{\pm0.1}$ | $72.7_{\pm0.1}$ | - | - | - | $87.5_{\pm0.0}$ | $88.2_{\pm0.0}$ | - | - | - |
| | MCDP | $71.4_{\pm0.1}$ | $73.0_{\pm0.0}$ | $70.8_{\pm0.1}$ | - | - | $87.7_{\pm0.0}$ | $88.2_{\pm0.0}$ | $86.8_{\pm0.2}$ | - | - |
| | DE | $71.2_{\pm\text{NA}}$ | $72.7_{\pm\text{NA}}$ | $71.6_{\pm\text{NA}}$ | - | - | $87.6_{\pm\text{NA}}$ | $88.2_{\pm\text{NA}}$ | $87.6_{\pm\text{NA}}$ | - | - |
| | EDL | $69.0_{\pm0.1}$ | $70.5_{\pm0.2}$ | $67.1_{\pm0.4}$ | $66.9_{\pm0.4}$ | $68.2_{\pm0.4}$ | $86.5_{\pm0.2}$ | $87.1_{\pm0.2}$ | $86.0_{\pm0.3}$ | $86.0_{\pm0.3}$ | $86.4_{\pm0.3}$ |
| | OE | $99.9_{\pm0.0}$ | $99.9_{\pm0.0}$ | - | - | - | $\mathbf{100.0_{\pm0.0}}$ | $\mathbf{100.0_{\pm0.0}}$ | - | - | - |
| | $DPN_{rev}$ | $98.9_{\pm0.1}$ | $99.1_{\pm0.0}$ | $99.2_{\pm0.1}$ | $99.2_{\pm0.1}$ | $99.2_{\pm0.0}$ | $99.7_{\pm0.0}$ | $99.8_{\pm0.0}$ | $99.8_{\pm0.0}$ | $99.8_{\pm0.0}$ | $99.8_{\pm0.0}$ |
| | $DPN^+$ | $99.9_{\pm0.0}$ | $99.9_{\pm0.0}$ | $99.9_{\pm0.0}$ | $99.9_{\pm0.0}$ | $99.9_{\pm0.0}$ | $\mathbf{100.0_{\pm0.0}}$ | $\mathbf{100.0_{\pm0.0}}$ | $\mathbf{100.0_{\pm0.0}}$ | $\mathbf{100.0_{\pm0.0}}$ | $\mathbf{100.0_{\pm0.0}}$ |
| | $DPN^-$ | $99.9_{\pm0.0}$ | $99.9_{\pm0.0}$ | $\mathbf{100.0_{\pm0.0}}$ | $\mathbf{100.0_{\pm0.0}}$ | $0.8_{\pm0.0}$ | $\mathbf{100.0_{\pm0.0}}$ | $\mathbf{100.0_{\pm0.0}}$ | $\mathbf{100.0_{\pm0.0}}$ | $\mathbf{100.0_{\pm0.0}}$ | $58.5_{\pm0.0}$ |
| Textures | Baseline | $70.9_{\pm0.2}$ | $73.3_{\pm0.2}$ | - | - | - | $53.8_{\pm0.4}$ | $57.5_{\pm0.5}$ | - | - | - |
| | MCDP | $70.3_{\pm0.2}$ | $72.6_{\pm0.3}$ | $63.6_{\pm0.2}$ | - | - | $53.1_{\pm0.3}$ | $56.5_{\pm0.5}$ | $41.6_{\pm0.2}$ | - | - |
| | DE | $71.1_{\pm\text{NA}}$ | $73.4_{\pm\text{NA}}$ | $76.2_{\pm\text{NA}}$ | - | - | $53.9_{\pm\text{NA}}$ | $57.7_{\pm\text{NA}}$ | $62.2_{\pm\text{NA}}$ | - | - |
| | EDL | $71.0_{\pm0.1}$ | $74.0_{\pm0.4}$ | $70.4_{\pm0.9}$ | $70.2_{\pm0.9}$ | $71.9_{\pm0.9}$ | $52.8_{\pm0.5}$ | $55.8_{\pm0.4}$ | $54.3_{\pm0.9}$ | $54.2_{\pm0.9}$ | $54.9_{\pm0.8}$ |
| | OE | $95.8_{\pm0.3}$ | $96.4_{\pm0.3}$ | - | - | - | $95.5_{\pm0.3}$ | $96.1_{\pm0.3}$ | - | - | - |
| | $DPN_{rev}$ | $90.9_{\pm0.3}$ | $91.9_{\pm0.3}$ | $91.2_{\pm0.6}$ | $90.6_{\pm0.6}$ | $92.6_{\pm0.3}$ | $89.9_{\pm0.3}$ | $91.3_{\pm0.3}$ | $90.7_{\pm0.6}$ | $90.2_{\pm0.6}$ | $92.3_{\pm0.3}$ |
| | $DPN^+$ | $96.5_{\pm0.1}$ | $97.1_{\pm0.0}$ | $98.4_{\pm0.0}$ | $98.4_{\pm0.0}$ | $98.2_{\pm0.0}$ | $96.2_{\pm0.0}$ | $96.9_{\pm0.0}$ | $97.9_{\pm0.0}$ | $97.9_{\pm0.0}$ | $97.8_{\pm0.0}$ |
| | $DPN^-$ | $95.8_{\pm0.1}$ | $96.8_{\pm0.1}$ | $\mathbf{98.7_{\pm0.1}}$ | $\mathbf{98.7_{\pm0.1}}$ | $19.3_{\pm0.4}$ | $95.4_{\pm0.0}$ | $96.5_{\pm0.0}$ | $\mathbf{98.5_{\pm0.1}}$ | $\mathbf{98.5_{\pm0.1}}$ | $35.7_{\pm0.3}$ |