[Reviews · NeurIPS 2020]

Review 1

Summary and Contributions: Post-rebuttal update: -------------------------------------- I thank the authors for the clarification, especially regarding the justification of the logistic-sigmoid approximation. I would suggest the authors use the approximation *only on* the sum of the exponent of the logits (and remove the discussion regarding the sum of the logits). Then, you can argue more theoretically---in addition to the explanation in the rebuttal---that for the interval (-inf, eps) with eps close enough to zero, the approximation error is very low, while for the interval [eps, inf), it yields the desired behavior (increasing just like the exponential function, but bounded). Under the assumption that the authors will incorporate the feedback above and do further polishing of the manuscript, I will, therefore, increase my score. -------------------------------------- This paper proposes a modification to Dirichlet prior networks (DPNs) which enables them to better distinguish "distributional uncertainty" (for detecting OOD data) and "data uncertainty" (for detecting in-distribution misclassifications). The key idea is to add terms in the DPN loss to maximize and minimize the Dirichlet's precision (the sum of alpha) for in-distribution and OOD losses, respectively. These modifications are based on the observation that the precision of a Dirichlet control the spread of its mass, and for OOD data, the mass should be concentrated in all corners of the probability simplex, while in the case of an in-distribution misclassification, the mass should be concentrated in the center of the simplex. Experiments show that the proposed method improves previous DPNs' OOD data detection performance and misclassifications detection.

Strengths: I like the idea of this paper: I think the proposed method is an elegant and novel way to distinguish between two types of uncertainty present in classification networks. Theoretically, the method is sound since it is straightforward exploitation of a particular, well-known property of the Dirichlet distribution. While it can be improved further, the empirical evaluation sufficiently validates the authors' claim. I think the proposed method can be very useful in practice since it provides a practitioner not only OOD detection but also misclassification detection mechanisms.

Weaknesses: The main weakness of this paper lies in the proposed regularizer, specifically in the approximation of the precision of a Dirichlet. Here, the authors proposes to approximate \alpha_0 = \sum_c z_c(x) with a bounded function 1/K \sum_c \sigma(z_c(x)) =: \alpha_0^*, where \sigma is the logistic sigmoid function. The problem here is: In what sense \alpha_0^* is a good approximation to \alpha_0? A quick check via plotting convinces me that this is not a good approximation. I think the authors need to do a much better job of motivating this approximation, especially since this is the core of the method, and hence the paper---if this approximation is wrong, then everything breaks down.

Correctness: - For an assessment of theoretical correctness, please see the previous section. - Methodologically, the method seems correct since it follows directly from a well-known property of the Dirichlet distribution. - The experiments follow the standard OOD detection experiment protocol, so I do not see anything wrong here.

Clarity: The paper is reasonably well written and conveys the motivation and idea nicely. Nevertheless, there are some errors and imprecision in the presentation: - Line 1-2: The claim that only DPN can model different uncertainty types is not true. There are methods like EDL (Sensoy et al., NeurIPS 2018) and the Laplace bridge (Hobbhahn et al., 2020) that also model the distribution of output distributions as Dirichlet, thus they should also be able to quantify similar uncertainty types just like DPN. - Line 54 etc.: p(\omega_c | x^*, \theta) is not a posterior (it is a likelihood in Bayesian inference). To see this more clearly: Does this distribution arises from the Bayes rule? Is it conditioned to the dataset D_in? - Line 61: The use of the word "variational" is imprecise since the author mentioned Langevin dynamics, which is an MCMC method. - Line 63: "the simplex", which simplex? The authors should write "the probability simplex" instead. - Line 68: "... it is even harder to ..." I think the problem is more about computational complexity, and not about finding an appropriate prior and inference algorithm. This is because a simple Gaussian prior along with simple Laplace approximation or variational Bayes has been known to yield good uncertainty estimates (see e.g. Ober & Rasmussen, 2019 and Kristiadi et al., ICML 2020). - Line 103: "point-estimate categorical distribution \mu"---\mu is just the parameter, and not the distribution itself. - Line 185: Eq. 4 is a softmax, not the cross-entropy loss. - Line 190: Which sigmoid function is used? Do the authors refer to the logistic function instead? - Line 203: What does "smaller modes" mean?

Relation to Prior Work: While the authors provide a sufficiently broad literature survey, they fail to discuss a very closely related method: the evidential deep learning (EDL, Sensoy et al., NeurIPS 2018), which as DPN, uses Dirichlet distribution to approximate the distribution of output distributions. The authors should discuss EDL in the related work section and empirically compare the proposed method with EDL. Additionally, the authors should also use the deep ensemble and a proper Bayesian neural network (not just MC-dropout) as baselines.

Reproducibility: Yes

Additional Feedback: Questions: - Could you explain where does the first equality in Eq. 12 come from? Suggestions: - Add a discussion regarding the choice of the logistic approximation (see the "Weaknesses" section for more detail). Also: a theoretical or numerical analysis on this approximation would make the paper much stronger. - Add comparison with EDL and deep ensemble. - Improve the clarity of the text and make it more precise. - Table 1: Please add show the corresponding accuracy. - Table 1, 2, 3: (i) It is much clearer to mark the best value *for each column* (ii) please increase the number of trials for computing the mean and std. dev. - Add EDL and deep ensemble baselines in the experiments. Minor comments: - Line 119: "... to each class ..." shouldn't it be "to each input x^*"? - Line 138: The digamma function should be introduced after Eq. 7, since it is not used in Eq. 6. - Line 148: Remove "r" in \beta \tilde{p}r. - Fig. 4, 5: Please make the text more legible, maybe make them span the page width? - Line 267: "... for sharp for ..." - Fig 5: Shouldn't "Confident predictions" rather be "Correct predictions"? - There are currently two references to VGG net ([1], [30]) To summarize: I really like the idea behind this paper. However, because of (i) the lack of justification on the core approximation used by the proposed method, (ii) the imprecision of the presentation, and (iii) the lack of important comparison with EDL, I vote for rejection. I would be more than happy to increase my score if the authors successfully addressed these three points.


Review 2

Summary and Contributions: The submission proposes to improve the uncertainty quantifying behaviour of Dirichlet Prior Networks (DPN) [1] by emphasising the difference in Dirichlet distributions for in-distribution-misclassified vs. OOD examples. In-distribution misclassified examples should produce flatter Dirichlet distributions towards the middle of the simplex, while OOD data should push away from the middle, since the confusion isn't about in-domain classes. This is achieved by placing a penalty on the logits of a DPN to push them to be negative for OOD examples. This, along with the entropy regulariser of [2], combine to encourage the distribution over predictive-distributions for OOD data to be a sharp Dirichlet along the edges of the simplex, which makes it easier to disambiguate the source of uncertainty. Experiments indicate that the difference in the two sources of uncertainties are well-captured. OOD detection performance is also improved over related baselines. [1] Predictive Uncertainty Estimation via Prior Networks, Malinin and Gales, 2018 [2] Deep Anomaly Detection with Outlier Exposure, Hendrycks et al., 2018

Strengths: To my knowledge, the proposed method is novel and derived from a principled framework about uncertainty quantification. Extensive experiments, with ablations and illustrative examples, have been conducted. Being able to distinguish the two causes for “confusing” data - OOD data vs. in-domain class confusion - can have practical significance, since the two situations might call for different interventions.

Weaknesses: Given that the major contribution of the paper is to improve detection of misclassified and OOD samples, it is a bit disappointing that the misclassification results in Table 2 and 8 are comparable at best with competing methods. While Table 3 and Fig 5 show that the combination of log(-diff. entropy) and MaxP allows distinguishing correct/incorrect/OOD examples, I’m not sure I noticed a formula/method for categorising an uncertain sample as being either in-domain-misclassified or OOD. In Fig 5, I’m curious what is the transform applied to the y-axis? Isn’t MaxP supposed to be bounded in [0,1]? L518 of the Appendix says: "In practice, since we do not know the characteristics of the OOD test examples, it may not be suitable to use a binary classifier for OOD detection tasks." This valid criticism also applies to the proposed method, since examples of outliers are used. Using binary classifiers has been shown to be very effective for highly practical tasks, for example, [3]. While the OOD detection scores are mostly near-perfect, can the authors comment about the reasoning in [4] where detecting dataset shift is not useful besides being likely easy-to-solve with trivial methods, than detecting “semantic” anomalies? For example, CIFAR-10 and STL-10 share quite a few of the same categories; yet STL-10 is treated as OOD for C10 in Table 1. [3] Reframing Unsupervised Machine Condition Monitoring as a Supervised Classification Task with Outlier-Exposed Classifiers, Paul Primus, 2020 [4] Detecting semantic anomalies, Ahmed et al., 2019 ————— Post rebuttal —————— Thanks for responding to some of my concerns. Perhaps the main results for misclassification in Table 2 should use AUROC and relegate AUPR to the Appendix, since base rates can differ. However, the results are still only comparable. Re. “detecting distributional shift across non-semantic factors can be useful for in-distribution generalization”: Improved performance at such tasks then implies that the final product, the classifier, gets more confused if non-semantic factors like contrast/image-compression/etc change even if the object is the same. And perhaps it does not get less confused if the non-semantic factors do not change but the object identity does. This is not the way we want our classifiers to behave. I believe the point of [4] is to disambiguate such problems. I'd encourage the authors to look into a less ambiguous test setup, such as [5] for example. Having read the rebuttal, I’ll retain my original rating. [5] Natural Adversarial Examples, Hendrycks et al., 2019

Correctness: As far as I can tell, the claims and method are correct, with correct empirical methodology (with the caveats mentioned above).

Clarity: The paper is clearly written and was easy to follow.

Relation to Prior Work: To my knowledge, the paper discusses prior work adequately.

Reproducibility: Yes

Additional Feedback:


Review 3

Summary and Contributions: The paper presents an improvement to the Dirichlet Prior Network (DPN) framework [5] to estimate uncertainties in deep models. In this framework, the prediction of a classification neural network with softmax activation output is interpreted as estimating the mean of a Dirichlet distribution, whose sharpness allows to measure predictive uncertainty and can be used to detect misclassified and Out-of-Distribution (OOD) examples at test-time. To obtain different spreads, in [5] KL-divergences with respect to a sharp and a flat target Dirichlet distribution for, respectively, in-domain data and OOD data are minimized at training-time. [16] proposed an improved version of DPNs. It uses a reversed KL-divergence as loss, which was shown to better handle cases of high data uncertainty: in such cases, a DPN should produce a uni-modal Dirichlet distribution placed in the right zone of the simplex between confusing classes; yet, the forward KL divergence from [5] actually results in a multi-modal distribution with a mode in each corner of the simplex. The submission further improves the loss for a DPN: the authors show how the choice of the target distributions for the reverse loss proposed in [16] still incurs the problem of flat output distribution for misclassified data, which in turn may affect OOD examples detection. The authors further show how simply changing the concentration parameters of the target Dirichlet distributions results in uncontrolled values of the concentration parameter for OOD examples. Hence, they propose a new loss, based on standard cross-entropy for the classification task plus an explicit precision regularizer aimed at controlling the sharpness of the output Dirichlet distribution. By carefully studying and selecting hyper-parameters of the loss, the authors force it to produce sharp multi-modal distributions for OOD examples, easing the task of telling them apart from in-domain data. A relatively large number of experiments validate the proposal.

Strengths: 2.1 - The paper performs a deep and non-trivial analysis on the effects of minimizing the loss proposed in [16], which, in spite of a not-always polished presentation (as discussed below), seems theoretically well-grounded to me. 2.2 - The contribution is novel. 2.3 - The contribution is relevant to the NeurIPS community: DPNs is a recently proposed framework to model predictive uncertainty without relying on ensembles, which was proposed and improved in recent NeurIPS submissions [5, 16]. 2.4 - I appreciate the effort the authors put into crafting examples and visualizations that give helpful intuitions to follow the formal and theoretical derivation. 2.5 - Experimental results on vision benchmarks are remarkably good for a method that offers a nice theoretical interpretation and allows to break down predictive uncertainty of a model into the standard model uncertainty and data uncertainty, and, peculiar to this framework, the distributional (or out-of-sample, OOD) uncertainty, without using ensembles.

Weaknesses: In order of importance: 3.1 - The main claim of the paper is that the Reverse-KL loss leads to indistinguishable distributions between OOD examples and misclassified examples. This is mainly justified through example 7 and the discussion at lines 147-152. However, equation 7 and its discussion reason on expectations on the training dataset and relative weights between different parts of the optimization objective: in other words, I don't think indistinguishable representations are deterministically implied by the theoretical analysis. Hence, the authors should validate the quantitative relevance of this problem also empirically, by showing this problem arises for thr RKL model in their synthetic experiment. Showing that the new loss performs better than the reverse KL loss is interesting, but there are too many differences between the two losses (e.g. the use of task-specific cross entropy term in the proposed loss) to ascribe such gain in performance exactly to the mitigation of this potential problem, which is instead assumed in the paper. 3.2 - Although results in terms of OOD detection are very high on an absolute scale, I think it would still help to assess thei relevance if at least one method from the recent literature is added to the comparisons in table 1 and 2. For instance, recent non-Bayesian approaches like [15] or [25] are cited but gently dismissed because they "cannot robustly determine the source of predictive uncertainty". I agree that the proposed framework is built on a sound theory and offers more flexibility when estimating predictive uncertainty. However, the authors are using also Maximum Softmax Probabilities as a total uncertainty measure in their experiments. So, it would be important to compare with at least one recent method, e.g. [15], which uses that same measure, to put the results on the paper in context. As it stands, the proposed approach is only compared with baselines and previous DPNs, which validates the improvement over previous DPN proposals, but is not helpful in assessing the performance of the framework on a relative scale with respect to the state of the art 3.3 In [16], the authors claim that one of the benefits of the reverse KL-divergence loss is that it allows DPNs (verbatim from the conclusions) "to be trained on more complex datasets with arbitrary numbers of classes.". As far as I can tell, this was due to the fact that the task-specific cross entropy term, used in [5], can be dropped from the loss when using the reverse divergence. The current submission reintroduces the CE term, but it does not comment on its effect on the feasibility of applying DPNs using the proposed loss on datasets with nowadays relatively large numbers of classes. 3.4 - Uncertainty metrics: the authors use the Mutual Information as a measure of uncertainty in figure 2. However, MI is defined with respect to two random variables and the authors don't indicate the second distribution they are comparing to. This makes it impossible to get a sense of what those MI values actually indicate. 3.5 - Discussion about equation 7 at lines 147-152 is mostly correct, but I find references to \beta being distributed among terms of the sum confusing: beta is a constant, common to all addends, that does not affect the optimization (apart from relative importance of the reverse cross entropy term wrt other losses, which is an effect not relevant to the discussion in the paragraph). To me, it makes more sense to think of the model prediction p(\omega_c | x) as changing between samples with low or high data uncertainty and affecting the optimization result. 3.6 - \bar{\beta} has not been explicitly defined, but its definition is important for the derivation in (7). I Had to look it up in [16] and assume the same definition was used. 3.7 - lines 153-166. This example doesn't add much to the paper in my opinion: it is obvious that the same precision can result in differently spread Dirichlet distributions. It is also wrongly placed, since it does not corroborate the reasoning on equation 7, as it is not obtained from optimizing the distribution parameters according to Eqn. 7. 3.8 Figure 4 is too small: I can't read the values on the legends nor I can see if, as claimed in the main text, "Max.P produces lower scores in the class overlapping regions." Please rearrange the text to make this figure have a reasonble size. 3.9 - lines 254-255 "It also demonstrates the difficulty of the existing non-DPN models to distinguish between data and distributional uncertainty". This claim seems unsubstantiated: section 5.1 trains and evaluates only DPNs, how can it demonstrate something about non-DPN models?

Correctness: The derivation of the equations which constitute the main contribution of the paper seems mostly correct to me (once the definition of \bar{\beta} is given). The empirical methodology seems also correct.

Clarity: The paper presentation needs to be improved. Notations is sometimes confusing: symbols are not always adeguately defined, and sometimes two symbols are used to refer to the same concept. For instance, thorught the text "pr" is used to denote probability, while "p" is used in the equations. A non-exhaustive list of typos or suggestions to improve follows: - Line 74: for each OOD distributions - > for each OOD distribution - Line 100: Dirichlet distribution -> Dirichlet distributions - Eq 1 uses D_in, from (2) on it becomes D - Line 113: given amount of large training data -> given large amount of training data - line 123: the class, c -> the class c - line 139: "where, \psi is the digamma function." -> there is no digamma function in (6). - line 147 and 173: eq 6 is eq 7 - line 148: factor, \beta. -> factor \beta. - line 161: "for than" -> "than" - line 161: more flatter and diverse -> flatter and more diverse - line 176: "for \tau :" -> "for \tau." - line 185: to models -> to model

Relation to Prior Work: The relation and differences to previous works on DPNs is clear and clearly discussed. Relevant prior works are cited and properly discussed.

Reproducibility: Yes

Additional Feedback: === POST-REBUTTAL UPDATE === I've read the rebuttal and the other reviews, and I retain my positive rating. The most important weaknesses I pointed out were successfully addressed in the rebuttal.


Review 4

Summary and Contributions: This paper further develops the previous study, the Dirichlet prior network (DPN) [1], which focuses on distinguishing misclassified in-distribution and out-of-distribution(OOD) samples. The paper emphasizes that, through the value of the Dirichlet distribution's precision, out-of-distribution, and misclassified samples can be further distinguished. It shows that high-valued precision has a sharp simplex distribution at the single corner while small precision sharpens the simplex distribution in all corners. In this regard, the author introduces a method of maximizing the precision of in-distribution samples and minimizing out-of-distribution samples' precision in the learning process to distinguish the distributions further. The proposed method consistently shows improvement among various OOD detection datasets. [1] Predictive Uncertainty Estimation via Prior Networks (Malinin et al., NeurIPS 2018)

Strengths: (1) Well motivated and intuitive: The author made good use of the characteristics of the Dirichlet distribution, making it very motivating, and proposed intuitive regularization. (2) Simple and effective: The proposed regularization is easy to use and has demonstrated consistent improvement of OOD detection performance in various datasets. Also, the author provided a well-designed ablation study, which proves the effectiveness of the proposed method.

Weaknesses: Main Questions (1) Question about the in-direct approach in equation (11). The goal is to maximize the precision \alpha_0; however, the author maximized the sum of the sigmoid(logits). Is there a reason for optimizing the in-direct variable rather than directly regularizing precisions? (e.g., empirical justification and intuition?) Due to this approximation, it is hard to see the direct connection between Dirichlet prior. It is more close to outlier exposure (OE) [1] + logit regularization. (2) Question about the misclassified samples. The main contribution of Dirichlet prior network (DPN) [2] is to separate misclassified sample uncertainty (high data uncertainty) and distribution uncertainty. Table 8 shows that DPN^{-} have better separation with misclassified samples and corrected samples. However, Table 2 shows no improvement in detecting such samples and sometimes have a significant drop. How can one interpret such results? (3) Question-related to Question #2. Is it possible to show the calibration error? (ECE [3] or root mean square calibration error[1]). Since the author's method maximizes the gap between in-distribution and out-of-distribution, the concern is that it might lead to over-confidence in in-distribution. Also, the proposed objective directly maximizes the logit value of the in-distribution sample. Therefore the model might lead to over-confidence. Minor Questions (4) Question about the out-of-domain set for training. The limitation of the DPN and outlier-exposure is that the out-of-domain sets are required for training [1, 3]. Is the performance dependent on the type of out-of-domain set? Since the CIFAR-10 and 100 are the subsets of 80 million tiny images dataset [4], they share similar characteristics. Can we use a dataset with highly different characteristics such as SVHN or Gaussian noise as out-of-domain when CIFAR is in-distribution set [5]? (5) Question about the detection score function. Is it possible to compute \alpha_0 for non-prior networks (e.g., OE)? The sum of exp(logit_c) can also be defined in such networks. [1] Deep Anomaly Detection with Outlier Exposure (Hendrycks et al., ICLR 2019) [2] Predictive Uncertainty Estimation via Prior Networks (Malinin et al., NeurIPS 2018) [3] On Calibration of Modern Neural Networks (Guo et al., ICML 2017) [4] https://groups.csail.mit.edu/vision/TinyImages/ [5] Training Confidence-calibrated Classifiers for Detecting Out-of-Distribution Samples (Lee et al., ICLR 2018) -------------- I read all of the author's rebuttal, and thank you for the detailed reply. 

I will raise one point from the first score. I mainly asked two questions, and one of them was addressed through the rebuttal, and the other was not fully sufficient. (1) The concern was about misclassification detection. First of all, RMS calibration error has shown that the proposed method indeed helped in-distribution classification uncertainty. Therefore, I agree with the effectiveness of the proposed method and will raise my score. (However, in the rebuttal line 18-23, it is difficult to agree that AUPR is not an ideal metric and AUROC is ​​a better choice. If so, I don't understand why AUPR was used as a compared metric.) (2) The other was the justification for the sigmoid function. As described in the paper, bounding the values (e.g., logits) can be fully understood, since the optimizing values are not bounded. However, it is still hard to understand why the logits have to be bounded rather than the precision. e.g., sigmoid \alpha_0
 For this reason, I will raise one point from the existing score.

Correctness: The claims are well motivated, and the method seems to be correct. However, as mentioned in "Weaknesses," some claims are questionable.

Clarity: The overall writing was easy to follow and shown intuitive examples for understanding. However, due to the inconsistency of notation in Section 4, some claim was hard to follow. Some examples are as follows: line 188 claims the regularization to maximize the sum of logits; however, the loss function maximizes the sum of the sigmoid(logit) Is P_r, pr, P, p the same probability? (page 4 line 138, equation 6, line 173, equation 8) \pshi is not defined in equation 6. line 199: \tilde{P}{out}, \lambda_{out} should be introduce after equation (10)

Relation to Prior Work: Yes. The proposed method significantly differs from previous contributions.

Reproducibility: Yes

Additional Feedback: There are two main reasons for rejection. First, the in-direct regularization for precision maximization is questionable. Second, the proposed method seems to show a tradeoff between out-of-distribution detection and misclassification detection. If the concerns are well justified, the paper will be more convincing and truly believe that the paper should deserve a better score. Minor feedback (1) Although the evaluation metrics (score) are commonly used, the explanation is in the 'Section 5.1 Synthetic dataset'. It would be better to have a description of the evaluation metric separately. (2) Ablation study of \lambda_{in, out} (section A.1) was convincing and have shown the effectiveness of the proposed method. It might be better to handle this section on the main text rather than the appendix.

[Author Response · NeurIPS 2020]

We thank the reviewers for their insightful comments. We first address the major concerns raised by the reviewers,
followed by their minor questions/ comments. We shall incorporate their suggestions into the paper.

**[R1 & R4] Justification of sigmoid (logistic) function in proposed**
**regularizer (Eq 9, 10).** By limiting logits, $z_c$ to values that are (ap-
proximately) greater than 5 for in-domain examples, and less than
-5 for OOD examples, we would have the desirable sharp uni-modal
or multi-modal Dirichlet distributions respectively, maximizing their
representation gaps (recall Fig 1; paper). Beyond these values, the
cross-entropy loss should be the dominant term in the loss function to
improve classification accuracy. The use of sigmoid function in our regularizer satisfies this condition by providing an
implicit upper (lower) bounds on the concentration parameters for in-domain (OOD) examples (see Fig(Rebuttal)(a)).

(a) $\frac{1}{K}\sum_c \text{sigmoid} z_c(\boldsymbol{x})$  (b) $\sum_c \exp z_c(\boldsymbol{x})$

Fig(Rebuttal): Growth of regularizers w.r.t logits.

In contrast, using the precision, $\alpha_0 = \sum_c \exp z_c(\boldsymbol{x})$ as the regularizer leads to large logit values for in-domain examples
(see Fig(Rebuttal)(b)). However, it makes the cross-entropy loss term negligible (Eq. 9), leading to degrading the
in-domain classification accuracy. Further $\exp z_c(\boldsymbol{x})$ is not a symmetric function. Hence, it does not equally constrain
the network to produce small fractional concentration parameters (i.e $\alpha_c = \exp z_c(\boldsymbol{x}) \to 0$) for OOD examples,
that leads to the desired multi-modal Dirichlet distributions (Fig 1d; paper). Moreover, in practice the choice of
$\sum_c \exp z_c(\boldsymbol{x})$ (or $\sum_c z_c(\boldsymbol{x})$) leads the training loss to NaN.

**[R2 & R4] "Sometimes have a significant drop" in misclassification detection Table 2.** Table 2 presents the AUPR
scores for misclassification detection. However, AUPR may not be an ideal metric for comparison, as it greatly depends
on the *base rates* i.e no. of misclassifications vs correct predictions (see accuracy vs. AUPR scores in Appendix Table
8) [1]. We instead recommend comparing the AUROC scores in Table 8 (appendix), where we achieve comparable
scores with the other non-ensemble based OOD models. Further, DPN is consistent with Bayesian ensemble techniques
(Eq 2, without marginalizing $\boldsymbol{\theta}$; Lines [107-115]), which would further improve the misclassification detection task.

Table (Rebuttal) gives the comparison of *root mean square calibration error (RMS)*
using the same experimental setup as Hendrycks et al [1]. We achieve simi-
lar performances as non-Bayesian OE [1], and better results for C100 and TIM.
Our proposed regularizer scales up (or down) the concentration parameters, $\alpha_c = $
$\exp z_c(\boldsymbol{x})$ for in-domain (OOD) examples, without disturbing their relative values i.e
$\exp z_c(\boldsymbol{x})/\sum_c \exp z_c(\boldsymbol{x})$ $\big(= p(\omega_c|\boldsymbol{x}^*, D)$; the predictive categorical$\big)$. Hence, it does not
lead to over-confidence for in-distribution examples.

|  | C10 | C100 | TIM |
|---|---|---|---|
| Baseline | $16.2_{\pm0.0}$ | $6.6_{\pm0.3}$ | $5.2_{\pm0.0}$ |
| MCDP | $15.7_{\pm0.1}$ | $6.7_{\pm0.0}$ | $5.3_{\pm0.2}$ |
| DE | $16.1_{\pm\text{NA}}$ | $6.8_{\pm\text{NA}}$ | $6.2_{\pm\text{NA}}$ |
| EDL | $15.5_{\pm0.1}$ | $10.1_{\pm0.4}$ | $10.3_{\pm0.4}$ |
| OE | $6.4_{\pm0.4}$ | $3.8_{\pm0.1}$ | $4.2_{\pm0.1}$ |
| $DPN_{rev}$ | $9.2_{\pm0.4}$ | $10.4_{\pm0.1}$ | $7.2_{\pm0.5}$ |
| $DPN^+$ | $\mathbf{6.3_{\pm0.3}}$ | $4.3_{\pm0.0}$ | $2.8_{\pm0.3}$ |
| $DPN^-$ | $6.5_{\pm0.2}$ | $\mathbf{3.5_{\pm0.1}}$ | $\mathbf{2.7_{\pm0.3}}$ |

Table(Rebuttal): Root mean square calibration error

Finally, our proposed maximizes the representation gap between in-domain and OODs
to *confidently determine the source of uncertainty* and improves the *OOD detection*
*performance*. Maximizing the gap between in-domain correct predictions and misclassifications is an important and
interesting problem for future research.

**Reviewer 1: Eq. 12**: First term is an expectation on joint dist. $\tilde{P}_T(\boldsymbol{x}, y)$ where $\boldsymbol{x}$ and $y$ are continuous and discrete
random variables. Denoting sigmoid (logistic) function as $\sigma$ (apply $p(x, y) = p(y|x)p(x)$ followed by rearranging):

$$\mathbb{E}_{\tilde{P}_T(\boldsymbol{x},y)}\Big[\sum_{c=1}^{K}\frac{-\lambda_T\sigma(z_c(\boldsymbol{x}))}{K}\Big] = \int_x\Big[\sum_y\Big[\sum_{c=1}^{K}\frac{-\lambda_T\sigma(z_c(\boldsymbol{x}))}{K}\Big]p(y|\boldsymbol{x})\Big]p(x)dx = \mathbb{E}_{P_T(\boldsymbol{x})}\Big[-\frac{\lambda_T}{K}\sum_{k=1}^{K}p(y=\omega_k|\boldsymbol{x})\Big[\sum_{c=1}^{K}\sigma(z_c(\boldsymbol{x}))\Big]\Big]$$

**Deep ensemble (DE)** results and **In-domain accuracy** are included in Appendix Table 8-12 and Table 8. We have also
included the results for **EDL** in Table (Rebuttal) and shall include their remaining results in our main paper.

**Reviewer 2: Fig 5:** We normalized the scores for better visualization.
Detecting **distributional shift across non-semantic factors** can be useful for in-distribution generalization on specific
domains by understanding the limitations of a classifier (Yarin Gal, Ph.D. thesis; 2016).

**Reviewer 3: Q 3.1** Appendix A.2 and B.1 provide additional ablation studies and results on the synthetic dataset for
Reverse-KL loss to further justify our claims. (Due to space constraints, we could not include them in the main paper.)
**Q 3.2** In Table 1 and 2, OE represents the non-Bayesian model (state-of-the-art) by Hendrycks et al [1].
**Q 3.3** Our C10, C100 and TIM tasks respectively uses CIFAR-10, CIFAR-100 and TinyImageNet with 10, 100 & 200
classes. Please refer to Appendix Table 7 where we present the experimental setup.

**Reviewer 4: Q4: Dataset with highly different characteristics** as OOD training set lead to poor OOD detection
performance. This is well-studied in (Lee et al., 2018; Hendrycks et al., 2019; Malinin et al., 2019).
**Q5:** $\sum_c \exp(z_c(\boldsymbol{x}))$ as a uncertainty measure for OE leads to poor OOD detection performances (in most of the cases)
as it does not control the absolute values of $\exp z_c(\boldsymbol{x})$ terms.

*Reference: [1] Deep Anomaly Detection with Outlier Exposure (Hendrycks et al., ICLR 2019)*

[Meta-Review · NeurIPS 2020]

This paper gives a modified loss for Dirichlet Prior Networks that is designed to help distinguish out-of-domain examples from in-domain but high-class-uncertainty ones. It is tested on a variety of small-image classification datasets. Three reviewers were mildly positive and one mildly negative so this is a borderline case. Neither the rebuttal nor a lengthy inter-reviewer discussion changed these views. There is a consensus that the approach is novel, interesting and potentially useful, but the negative reviewer requested further justification and several of the reviewers point out relevant related work that needs to be discussed. Overall, the AC agrees that the idea is elegant and novel enough to appear in NeurIPS. For the final version of the paper, the authors should address the two main concerns of R4, discuss some of the missing references and strengthen the justification of the various design choices. Also, the tables and images in the text are uncomfortably cramped so some of the less-important material from the current paper may need to be moved to the supplementary material.